# Neuroprotective Effects of VEGF-B in a Murine Model of Aggressive Neuronal Loss with Childhood Onset

**DOI:** 10.3390/ijms26020538

**Published:** 2025-01-10

**Authors:** Laura Pérez-Revuelta, David Pérez-Boyero, Ester Pérez-Martín, Valeria Lorena Cabedo, Pablo González Téllez de Meneses, Eduardo Weruaga, David Díaz, José Ramón Alonso

**Affiliations:** 1Laboratory of Neuronal Plasticity and Neurorepair, Institute of Neuroscience of Castile and Leon (INCyL), Universidad de Salamanca, 37007 Salamanca, Spain; laura.perez-revuelta@uk-koeln.de (L.P.-R.);; 2Institute of Biomedical Research of Salamanca (IBSAL), 37007 Salamanca, Spain; 3Neuroscience Innovative Technologies, Neurostech, 33428 Llanera, Spain; 4Instituto de Investigación Sanitaria del Principado de Asturias (ISPA), Intervenciones Traslacionales para la Salud, 33011 Oviedo, Spain

**Keywords:** neurodegeneration, neurotherapeutics, PCD mouse, cerebellum, Purkinje cells, neurotrophic factors

## Abstract

In recent decades, the scientific community has faced a major challenge in the search for new therapies that can slow down or alleviate the process of neuronal death that accompanies neurodegenerative diseases. This study aimed to identify an effective therapy using neurotrophic factors to delay the rapid and aggressive cerebellar degeneration experienced by the Purkinje Cell Degeneration (PCD) mouse, a model of childhood-onset neurodegeneration with cerebellar atrophy (CONDCA). Initially, we analyzed the changes in the expression of several neurotrophic factors related to the degenerative process itself, identifying changes in insulin-like growth factor 1 (IGF-1) and Vascular Endothelial Growth Factor B (VEGF-B) in the affected animals. Then, we administered pharmacological treatments using human recombinant IGF-1 (rhIGF-1) or VEGF-B (rhVEGF-B) proteins, considering their temporal variations during the degenerative process. The effects of these treatments on motor, cognitive, and social behavior, as well as on cerebellar destructuration were analyzed. Whereas treatment with rhIGF-1 did not demonstrate any neuroprotective effect, rhVEGF-B administration at moderate dosages stopped the process of neuronal death and restored motor, cognitive, and social functions altered in PCD mice (and CONDCA patients). However, increasing the frequency of rhVEGF-B administration had a detrimental effect on Purkinje cell survival, suggesting an inverted U-shaped dose–response curve of this substance. Additionally, we demonstrate that this neuroprotective effect was achieved through a partial inhibition or delay of apoptosis. These findings provide strong evidence supporting the use of rhVEGF-B as a pharmacological agent to limit severe cerebellar neurodegenerative processes.

## 1. Introduction

Neurodegenerative diseases are a group of disorders of the central nervous system (CNS) characterized by the progressive loss of neurons, and that may affect different regions. Although these diseases are more prevalent in adulthood, they can also manifest earlier, even in childhood, resulting in severe consequences caused by their impact on crucial phases of children’s physical, cognitive and emotional development. Recent studies have identified a new neurodegenerative disease called childhood-onset neurodegeneration with cerebellar atrophy (CONDCA), which affects the Purkinje cells of the cerebellum and causes severe ataxia from a very early age [1,2,3,4]. This disease results from the loss of function of the protein CCP1 (also known as AGTPBP1 or NNA1), which acts as a cytosolic carboxypeptidase. More precisely, impairments in this protein cause a destabilization of the cytoskeleton, amongst other cellular symptoms, that finally leads to neuronal death [3,5,6]. Patients that suffer from it present the virtual disappearance of Purkinje cells, which leads to a dramatic cerebellar shrinkage [1,2,3]. Fortunately, there is an animal model that exhibits both a mutation and a neurodegenerative process similar to those observed in CONDCA patients, the Purkinje Cell Degeneration (PCD) mutant mouse [7,8,9,10].

The effect of the PCD mutation is the postnatal and selective degeneration of certain neuronal populations in which the Ccp1 gene is normally highly expressed (i.e., the Purkinje cells of the cerebellum, as the most striking symptom [10]). The neurodegenerative process of Purkinje cells can be divided into two steps: (1) the pre-degenerative stage from postnatal day 15 (P15) to P18, in which nuclear, cytological, and morphological changes are detected [6,11,12,13,14]; and (2) the degenerative stage per se, from P18 onwards, when Purkinje cell death is evident, with only a few Purkinje neurons remaining in lobe X of the cerebellar vermis at P40 [6,7,10,12,13,15,16,17]. This neuronal death produces a severe cerebellar ataxia—the most representative phenotypic feature of PCD mutant mice—and other cognitive and affective impairments, from the third week of postnatal life onwards [6,15,18,19,20]. Considering these premises, the PCD mouse has been extensively used to investigate new neuroprotective strategies to prevent –or at least attenuate– neurodegeneration [20,21,22,23]. The main experimental approaches are aimed at reversing or preventing the aggressive cerebellar degeneration of this mutant model of CONDCA.

Neurotrophic factors are a group of secreted molecules that play a critical role in the development, differentiation, proliferation, and survival of CNS. They belong to several different molecular families, but they all have in common the fact that they are required for the proper functioning of neurons, stimulating cell signaling and axon and dendrite growth [24]. Therefore, therapies comprising neurotrophic factors constitute a promising area of research for the treatment of neurodegenerative diseases [25], as they can improve cognitive and motor function and reduce cell death in animal models of diseases such as Alzheimer’s or Parkinson’s [26,27]. However, both the administration pattern and the effectiveness of these therapies vary among the different neurodegenerative diseases [28,29,30], so it is essential to identify the most appropriate neurotrophic factor to be applied for obtaining putative benefits in a particular condition. Hence, the objective of this work was to examine the effect of the treatment with neurotrophic factors on cerebellar degeneration and neurobehavioral defects of PCD mice, as a model of CONDCA. To this end, we analyzed the expression of four molecules with high neuroprotective activity, in both the cerebellum and plasma of PCD mice: insulin-like growth factor 1 (IGF-1), brain-derived neurotrophic factor (BDNF), Vascular Endothelial Growth Factor A (VEGF-A), and Vascular Endothelial Growth Factor B (VEGF-B). Then, we studied the effect of the treatment with those neurotropic factors altered by PCD mutation by analyzing the survival of Purkinje cells and the motor, cognitive and social functions, impaired in this animal model.

## 2. Results

### 2.1. Changes in Gene and Protein Expression of Neurotrophic Factors

We first analyzed the gene expression of Igf-1, Bdnf, Vegf-A, and Vegf-B in the cerebellum of WT and PCD mice throughout the neurodegenerative process of the PCD mutant mouse: P10, P15, P20, P25, P30, and P40. In addition, we studied the production of the corresponding proteins in both cerebellar tissue and blood plasma at the key ages of the neurodegenerative process, P15 and P25 (pre-neurodegeneration and ongoing degeneration, respectively), and at those ages when we observed changes at the gene level. Our results revealed no differences between experimental groups in either gene or protein levels for BDNF (Figure 1A–C) and VEGF-A (Figure 1D–F). By contrast, significant differences were observed in the expression of IGF-1 (Figure 1G–I) and VEGF-B (Figure 1J–L) at both gene and protein levels.

In the case of IGF-1, we observed changes in PCD animals at P25 and P40 compared to WT mice. At P25 we saw an increased gene expression in mutants (Figure 1G; *p* = 0.010) corresponding to an increased protein production in the cerebellum of PCD mice (Figure 1H; *p* = 0.029), but not in blood plasma, where we observed a reduction in IGF-1 (Figure 1I; *p* > 0.05). Conversely, at P40 we noticed a somehow opposite situation: despite an increased gene expression in the cerebellum (Figure 1G; *p* = 0.016), the protein level of IGF-1 remained similar to the WT mice one (Figure 1H; *p* > 0.05), while it increased in the blood plasma (Figure 1I; *p* = 0.016).

For VEGF-B, we found fluctuations at both P15 and P20 in mutant animals. At P15, gene expression increased in PCD mice (Figure 1J; *p* = 0.016), which also corresponded to an increase in protein production in the cerebellum (Figure 1K; *p* = 0.042), but not in the blood plasma (Figure 1L; *p* > 0.05). On the other hand, at P20 we found a reduction in gene expression in the cerebellum (Figure 1J; *p* = 0.004) and in protein levels in the blood plasma in mutant mice (Figure 1L; *p* = 0.042), without changes in protein production in the cerebellar tissue (Figure 1K; *p* > 0.05).

In sum, our observations revealed fluctuations in both gene and protein expressions of IGF-1 and VEGF-B induced by the PCD mutation. These changes seem dependent on the time-course of degeneration. VEGF-B is associated with early stages: P15, at the start of pre-neurodegeneration, and P20, when Purkinje cells begin to die. By contrast, IGF-1 levels vary during advanced degeneration: at P25, when a massive death of Purkinje cells is detected, and at P40, when cerebellar neurodegeneration is complete, with only a few Purkinje cells surviving.

Based on these data, we developed an experimental design for administering rhIGF-1 or rhVEGF-B, based on their genotype-specific expression variations. Treatment was initiated five days before the onset of these variations, specifically at P20 for rhIGF-1 and at P10 for rhVEGF-B, and lasted 10 days. For both treatments, we used a sham group of NaCl-treated PCD animals (PCD-NaCl) to test the effect of continuous injections, since their repetition may cause severe stress due to immobilization, cumulative irritant effects, or needle damage [31].

### 2.2. Expression of IGF-1 and VEGF-B Receptors in PCD Mice 

Before administering rhIGF1 and rhVEGF-B, we set out to determine whether the IGF1 receptor (IGF1R) and the VEGF-B receptor (VEGFR1) were expressed in Purkinje cells at the time of treatment. To achieve this, we examined VEGFR1 expression at P15 and IGF1R at P25 in both WT and PCD mice.

Due to the diffuse expression and distribution of these receptors, their analysis was qualitatively performed. Both IGF1R (Appendix A) and VEGFR1 (Appendix A) were expressed in all cerebellar layers, especially in the Purkinje cell layer; however, no qualitative differences were observed between genotypes (Appendix A).

Therefore, we confirmed that IGF-1 and VEGF-B receptors are present in the cerebellum of both WT and PCD mice during the ages analyzed in this work. Thus, rhIGF1 and rhVEGF-B are potentially suitable molecules for counteracting cerebellar neuronal loss.

### 2.3. rhIGF-1 Treatment

Treatment with rhIGF-1 was performed daily from P20 to P30, and its effect on motor, cognitive and social behavior, and Purkinje cell survival was evaluated.

#### 2.3.1. The Treatment with rhIGF-1 Does Not Improve Memory Impairments of PCD Mice

Starting with the Novel Object Recognition (NOR) test, we found the same results at both P25 and P30 concerning the percentage of interaction time for each object. While WT mice spent a greater percentage of time interacting with the new object (Figure 2A,B; *p* P25 = 0.004; *p* P30 = 0.001), no significant differences were observed between the time spent investigating the new object and the familiar one in PCD, PCD-NaCl, and PCD treated with rhIGF1 (PCD-rhIGF1) mice (Figure 2A,B; Appendix A; *p* > 0.05). Additionally, as a complement to the above findings, we analyzed the DI.

Our results indicated that WT mice exhibited a positive discrimination index (DI) at all tested ages, highlighting their proficient ability to differentiate between familiar and novel objects and thus, indicating a well-functioning recognition memory. Conversely, the PCD, PCD-NaCl, and PCD-rhIGF1 mice demonstrated a DI close to 0 at all ages (Figure 2C). Kruskal–Wallis statistical analysis, followed by a post hoc test that clusters data in statistically different homogeneous subgroups, showed differences in WT mice compared to the three PCD groups (PCD, PCD-NaCl, PCD-rhIGF1) at all ages analyzed (*p* < 0.001; Figure 2C). Finally, we analyzed the total number of visits to both objects to exclude any potential influence of ataxia in PCD mice. This study was conducted at P30, when PCD mice typically exhibit a severe ataxia, but no significant differences were detected between experimental groups, thus validating the former results (Figure 2D; *p* > 0.05).

As a result, our findings demonstrated that administering rhIGF-1 does not lead to an improvement in recognition memory in PCD mice. Additionally, we did not observe any differences between PCD and PCD-NaCl mice, which indicates that the continuous injection methodology had no impact on these results.

#### 2.3.2. rhIGF-1 Administration Does Not Normalize Social Behavior

Our results demonstrated that WT mice had a preference for interacting with another animal at all ages examined, as significant differences in the time spent with an animal compared to an object were found (Figure 2E,F; *p* P25 = 0.015; *p* P30 = 0.009). Conversely, any preference was not detected in PCD, PCD-NaCl, and PCD-rhIGF1 mice, at any of the ages examined (Figure 2E,F; *p* > 0.05). The analysis of the SI confirmed the previous results, since WT mice scored positively at all ages tested, indicating social preference (Figure 2G). By contrast, PCD, PCD-NaCl, and PCD-rhIGF1 mice showed values close to 0 (or even lower) at both P25 and P30, indicating a complete lack of social preference (Figure 2G). Kruskal–Wallis and subsequent post hoc analyses revealed two clearly distinct homogeneous subgroups: WT mice vs. all PCD animals, regardless of the treatment (Figure 2G). Furthermore, we analyzed the overall number of visits to each room at P30 (Figure 2H) to assess the impact of ataxia. Consistent with our previous findings, no significant disparity in visits to either the animal or the object was observed, thus excluding the influence of the ataxia in the mutant mice.

In sum, similarly to the previous experiment, we excluded any impact of both rhIGF-1 treatment and daily administration on the social behavior of PCD mice.

#### 2.3.3. Effect of Treatment on Motor Coordination

The repeated measures one-way ANOVA statistical analysis indicated an interaction between the “day of testing” and “experimental group” factors (*p* = 0.001), suggesting that both factors simultaneously influenced the observed outcomes. This is evident in the different performance curves when comparing genotypes (Figure 2I): while WT mice experienced a progressive improvement in their motor behavior (an extended latency to fall; *p* = 0.01), PCD, PCD-NaCl, and PCD-IGF1 mice displayed impaired motor coordination from P25 onwards, with latency values dropping close to 0 at P30 (*p* < 0.001 for both ages and experimental groups). The absence of learning and decline in motor performance with age in the PCD experimental groups are attributed to cerebellar degeneration. To further analyze the data, and to obtain the maximum information about the effect of treatment, we analyzed possible differences between experimental groups at each age of testing using the non-parametric Kruskal–Wallis test. The results showed significant differences between WT and all PCD mice regardless of treatment (Figure 2I; *p* < 0.01 for both age and PCD experimental groups).

Hence, we can state that rhIGF-1 treatment does not enhance or restore the motor performance of PCD mice.

#### 2.3.4. rhIGF-1 Treatment Does Not Prevent the Purkinje Cell Death in PCD Mice

The analysis of Purkinje cell density demonstrated that WT mice had higher cell density compared to PCD animals independent of treatment (Figure 3A–D). The Kruskal–Wallis non-parametric test indicated significant differences between WT and PCD groups (Figure 3E; *p* WT-PCD = 0.002, *p* WT-PCD-NaCl = 0.002; *p* WT-PCD-rhIGF1 = 0.035). Since the large disparity of the data between WT and PCD mice could blur possible treatment results amongst mutant animals, a new statistical analysis was performed without considering WT mice (Figure 3F). However, we did not observe significant differences between PCD and PCD mice treated with rhIGF-1 or NaCl (Figure 3F; *p* > 0.05).

Thus, rhIGF-1 treatment does not appear to have a neuroprotective effect on Purkinje cell density in the cerebellum of PCD mice. Additionally, due to the absence of differences between untreated mutants and NaCl-treated mice, we discarded any effect related to daily drug administration.

Considering the aforementioned data, we can affirm that daily rhIGF1 treatment from P20 to P30 does not exert any neuroprotective or beneficial effect on the PCD mutant animals, according to the used conditions and tests. Consequently, further analyses using this neurotrophic factor were not conducted.

### 2.4. rhVEGF-B Treatment

The administration of rhVEGF-B in PCD mice (PCD-rhVEGFB animals) every other day started at P10, five days before the onset of pre-neurodegenerative changes, and continued for 10 days until P20. In addition, we used a new set of PCD animals treated with NaCl following the same administration pattern to assess possible effects of these two day-separated injections at an early age [31].

#### 2.4.1. rhVEGF-B Treatment Normalizes Memory Impairment

Concerning the exploration time of familiar and novel objects, we found that at P17, all experimental groups showed a comparable behavior, as they spent most of the time exploring the novel object (Figure 4A; *p* WT = 0.001; *p* PCD = 0.003; *p* PCD-NaCl < 0.001; *p* PCD-rhVEGFB < 0.001). Conversely, at P25 and P30, PCD and PCD-NaCl animals showed no preference, as they did not present any significant difference between the time spent exploring the novel or the familiar object (Figure 4B,C; *p* > 0.05). In contrast, at both P25 and P30, PCD-rhVEGFB mice showed a preference for interaction with the novel object, having a similar behavior to WT mice (Figure 4B,C; Appendix A; P25: *p* WT = 0.004; *p* PCD-rhVEGFB = 0.001; P30: *p* WT = 0.002; *p* PCD-rhVEGFB = 0.003). Then, we assessed the DI (Figure 4D). At P17, all the experimental groups displayed a correct ability to differentiate between familiar and novel objects, thereby having a clearly positive DI, without statistical differences amongst them (*p* < 0.001). However, the scenario changed at P25 and P30, as PCD and PCD-NaCl mice exhibited values close to 0, indicating no preference for any object. In contrast, PCD-rhVEGFB mice consistently displayed positive values, which closely resembled those observed in the WT mice. Statistical analyses were conducted using the Kruskal–Wallis test followed by post hoc testing to discriminate homogenous subgroups. Our findings showed that at P17, no significant differences were found between the experimental groups (*p* > 0.05; Figure 4D). However, data could be evidently separated into two distinct subgroups at P25 and P30 (*p* < 0.001). At both ages, the first group comprised WT and PCD-rhVEGFB mice, with positive values indicating their ability to discriminate between objects. The second group consisted of PCD and PCD-NaCl, with values close to 0 and lacking discriminatory capacity (Figure 4D). After analyzing the number of visits to both objects, we did not find any significant difference between the four experimental groups (*p* > 0.05), despite the fact that at this age PCD mice already show ataxia due to cerebellar degeneration (Figure 4E).

Therefore, rhVEGF-B treatment normalizes recognition memory in PCD mutant mice, resembling WT mice at all ages tested.

#### 2.4.2. Normal Social Behavior Is Maintained After rhVEGF-B Administration

Our findings indicated that WT mice exhibited a social preference at all ages tested (Figure 4F–H and Appendix A; *p* P17 = 0.000; *p* P25 = 0.006; *p* P30 = 0.004). Conversely, both PCD and PCD-NaCl mice did not show a preference at any of the ages analyzed, comparing the visits to the other animal or the object (*p* > 0.05). Nevertheless, rhVEGF-B-treated mice spent more time interacting with the animal than the object, displaying similar social preference to WT mice at all ages investigated (Figure 4F–H; *p* P17 = 0.000; *p* P25 = 0.005; *p* P30 = 0.001). Regarding the sociability index (SI), PCD and PCD-NaCl mice exhibited values nearing 0 at all ages analyzed (Figure 4I). In contrast, SI values were continually positive in WT and PCD-rhVEGFB mice, leading to the identification of two distinct homogeneous groups (Kruskal–Wallis analysis and corresponding post hoc tests; *p* < 0.001): WT and PCD-rhVEGFB mice, and PCD and PCD-NaCl mice.

According to the analysis of the number of visits to either the mouse or the object, no differences were detected between the experimental groups. Therefore, ataxia of mutant mice did not affect the performance of this test (Figure 4J; *p* > 0.05).

Similarly to recognition memory, treatment with rhVEGF-B prevented the social impairments observed in PCD mice, indicating a neuroprotective effect of this substance at the memory and social levels.

#### 2.4.3. Effect of Treatment on Motor Coordination

The repeated measures one-way ANOVA test showed an interaction between the “day of testing” and the “experimental group” factors (*p* < 0.001). Similarly to rhIGF-1 treatment, this interaction resulted from the data discrepancy between genotypes. Initially, WT mice improved their performance with age by increasing the latency to fall, probably due to learning (Figure 4K; *p* < 0.001). However, PCD, PCD-NaCl, and PCD-rhVEGFB mice did not display any improvement with age, but a decrease in the latency to fall, presumably due to the progressive cerebellar neurodegeneration (Figure 4K; *p* PCD < 0.001; *p* PCD-NaCl = 0.013; *p* PCD-rhVEGFB = 0.021).

We conducted a statistical analysis within each day of testing in order to further evaluate the differences between experimental groups at this level. The Kruskal–Wallis test showed that the motor behavior varied between genotypes at all ages. Firstly, the performance of the PCD and PCD-NaCl groups differed from WT mice from P17 onwards, with significant differences observed at all analyzed ages (Figure 4K; *p* < 0.001 for both experimental groups and ages with respect to WT). However, the motor coordination of rhVEGF-B-treated mutants was similar to WT mice at P17 (*p* > 0.05), but different to PCD mice (Figure 4K; *p* > 0.001). From P25 onwards, a decline was observed in the motor performance of PCD-rhVEGFB mice, showing statistical differences with WT animals (Figure 4K; *p* P25 = 0.016; *p* P30 = 0.001, *p* value PCD-VEGFB respect to WT mice). Nevertheless, PCD-rhVEGFB animals performed better in the Rotarod test than untreated mutant mice at P25 and P30 (Figure 4K), showing significant differences (*p* P25 = 0.023; *p* P30 = 0.002; *p* values with respect to PCD mice). On the other hand, we did not observe any differences between PCD and PCD-NaCl mice (*p* > 0.05), so it seems that there is no effect of this type of administration on motor behavior (Figure 4K).

Similarly to the other cerebellar functions tested here, rhVEGF-B treatment improved motor coordination, at least partially, as rhVEGF-B-treated mice generally showed results that were intermediate between those obtained for WT and PCD mice.

#### 2.4.4. rhVEGF-B Administration Decreases Purkinje Cell Death

##### Analyses at P30

To assess whether the improved behavior of PCD-rhVEGF-B animals was due to an effect at the histological level, Purkinje cell survival was quantified in all experimental groups at P30 (Figure 5A–D). Our results demonstrated a significant reduction in cell density for all PCD experimental groups, compared to the WT mice (Figure 5E; *p* PCD = 0.010; *p* PCD-NaCl = 0.001; *p* PCD-rhVEGFB = 0.008; *p* value compared to WT), along with an improvement in the PC morphology (Appendix A). Despite this, the PCD-rhVEGFB mice exhibited a slightly higher cell density, compared to untreated PCD (*p* = 0.009) and PCD-NaCl groups (Figure 5E; *p* = 0.006).

According to these results, we conducted an analysis of apoptosis (Figure 6). We quantified the linear density of TUNEL-positive cells in the Purkinje cell layer in relation to their length. The cells included in the quantification exhibited a typical morphology comprising small size, disorganized nucleus and TUNEL-positive labeling (Figure 6A–C). Since this administration pattern (injections every other day) showed no effect in PCD-NaCl mice in any of the above cases, we decided to exclude this experimental group from further analyses, to optimize resources and minimize the use of animals. The Kruskal–Wallis non-parametric test revealed a significant higher density of TUNEL-positive cells in PCD-rhVEGFB mice compared to both WT and PCD groups (Figure 6G; *p* < 0.001 in both cases). In addition, we observed no differences between the untreated PCD and WT mice (*p* > 0.05).

Altogether, these findings suggest that the administration of rhVEGF-B is causing a partially neuroprotective effect on Purkinje cells. Our observations show an intermediate cell density between WT and PCD mice at P30 that positively correlates with improved motor, cognitive, and social behavior. However, we also observed an increase in apoptotic cell density in mutants treated with rhVEGF-B, which might be explained by the higher Purkinje cell density of this group, as these neurons start to die later than those in untreated PCD mice. To investigate this hypothesis of a delay in Purkinje cell death and thus to discern whether rhVEGF-B provides greater neuroprotection at earlier stages or only a partial protection, we replicated these histological analyses at P25.

##### Analyses at P25

Quantification of Purkinje cell survival was performed in WT, PCD, and PCD-rhVEGFB mice (Figure 7A–D). As we did not find any differences due to continuous administration every other day at P30, we did not use the PCD-NaCl experimental group at this age, as previously stated. The non-parametric Kruskal–Wallis test showed a reduction in Purkinje cell density throughout the cerebellum in PCD mice compared to WT (Figure 7D; *p* = 0.002) and PCD-rhVEGFB animals (Figure 7D; *p* = 0.035). Furthermore, we did not find significant differences between WT and PCD-rhVEGFB mice (Figure 7D; *p* > 0.05), indicating that rhVEGF-B treatment completely prevents Purkinje cell death, at least up to P25.

Then, we conducted an analysis of apoptosis in the cerebellum of the same groups (Figure 7E–H). The Kruskal–Wallis test and corresponding post hoc analyses indicated significant differences solely between WT and untreated PCD animals (Figure 7D; *p* < 0.001). However, we did not observe differences between WT and PCD-rhVEGFB mice or between PCD and PCD-rhVEGFB animals (Figure 7H; *p* > 0.05). In addition, we performed an active caspase-3 analysis in the aforementioned experimental groups, and we found a qualitative increase in caspase-positive Purkinje cells in PCD-rhVEGFB mice compared to WT and PCD mice (Appendix A). Since caspase-3 is a marker for the onset of apoptosis and TUNEL identifies later stages, we observed the beginning of the cell death process at P25, culminating at P30 with a significant increase in TUNEL-positive cells. These findings support our hypothesis of delayed Purkinje cell death after rhVEGF-B treatment.

These findings suggest that rhVEGF-B treatment exerts a neuroprotective effect on Purkinje cells, as we observed a similar cell density to WT, which is associated with an improvement in cerebellar functions. However, the treatment partially attenuates or delays DNA damage and cell death, as we observed in PCD-rhVEGFB an intermediate number of TUNEL-positive cells between WT and PCD animals. In sum, we verified the hypothesis that rhVEGF-B treatment delays Purkinje cell death with a maximum effect at P25, which starts to decline around P30.

### 2.5. Treatment with rhVEGF-B Does Not Lead to Improvements in Skeletal Muscle

Cerebellar ataxias are not only associated with primary neuronal damage, but also cause secondary muscle atrophy [32]. Therefore, the functional improvement of treated mice could be based not only on the attenuation of Purkinje cell death, but may also be due to effects outside the encephalon, as has been observed in previous studies from our laboratory [19]. Consequently, we analyzed the skeletal muscle from WT, PCD, and PCD-rhVEGFB mice at P30 (Appendix A).

The Kruskal–Wallis statistical test showed no differences between the experimental groups, either in a longitudinal section (Appendix A), considering the major (Appendix A) or minor axis (Appendix A), or in a transverse section of muscle (Appendix A; *p* > 0.05). Therefore, the improvement in motor test performance does not appear to be due to an effect of rhVEGF-B on skeletal muscle, but almost exclusively due to an increased neuronal survival in the cerebellum.

### 2.6. Chronic Administration of rhVEGF-B Did Not Prevent Purkinje Cell Death

Given the neuroprotective effects of rhVEGF-B injected every other day, we wanted to test whether an increase in the frequency of administration could improve the results obtained. Thus, we decided to treat a new group of PCD animals with rhVEGF-B daily from P10 to P20, and we investigated the impact of this reinforced administration on cognitive, social, and motor behaviors as before.

Starting with the NOR test (Figure 8A–D), we observed similar behavior in all animal groups at P17, as they spent most of the time interacting with the novel object (Figure 8A; *p* WT < 0.001; *p* PCD = 0.004; *p* PCD-NaCl < 0.001; *p* PCD-rhVEGFB < 0.001). At P25 and P30, WT mice also demonstrated a preference for interacting with the novel object (Figure 8B,C; *p* P25 = 0.009; *p* P30 < 0.01). However, regarding all PCD groups, there were no significant differences between the time spent exploring the novel and the familiar object (Figure 8B,C; *p* > 0.05). Regarding DI, at P17 all experimental groups displayed a positive index, indicating a correct discrimination ability. However, from P25 onwards, the DI values of the different PCD groups—either treated or untreated—showed values close to zero or even negative, indicating an impaired recognition memory. The analysis of the overall number of visits to both objects at P30 indicated no disparities among experimental groups (*p* > 0.05), so we can infer again that the test performance had not been influenced by cerebellar ataxia (Figure 8D). Thus, rhVEGF-B administered daily did not enhance the recognition memory of mutant animals.

Next, social behavior was examined with the social preference test (Figure 8E–H). As in previous tests, WT mice exhibited a social preference at all ages evaluated (Figure 8E–G; *p* P17 = 0.029; *p* P25 = 0.011; *p* P30 = 0.002), while all PCD groups did not show social preference, and no differences were found between them at any of the ages assessed (Figure 8E–G; *p* > 0.05). Additionally, the SI analysis revealed that only WT mice exhibited positive values, whereas the other experimental groups displayed values approximately equal to zero or below (Figure 8H). Moreover, the analysis of the number of total visits to the mouse and the object at P30 also did not show differences (*p* > 0.05), indicating that cerebellar ataxia does not affect this test performance (Figure 8H). Hence, daily administration of rhVEGF-B did not improve social behavior in PCD mice.

The effects of continuous treatment on motor function were also evaluated at P17, P25, and P30 (Figure 8I). The repeated measures one-way ANOVA test once more demonstrated an interaction between the “day of testing” and “experimental group” factors (*p* < 0.001). Again, this interaction resulted from the data discrepancy between genotypes: while WT mice displayed a progressive improvement in motor behavior (*p* < 0.001), PCD experimental groups faced a decline in their performance, exhibiting values close to 0 at P30 (*p* < 0.001 for all PCD groups). Next, we conducted statistical analysis to compare the experimental groups within each age group. Consistent with our observations in previous behavioral tests, significant differences between the WT and PCD groups were detected (*p* < 0.001 for all ages and PCD groups compared to WT animals), without differences between PCD experimental groups (*p* > 0.05).

In sum, based on these data, it can be concluded that administering rhVEGF-B daily fails to enhance the motor, cognitive, and social abilities of PCD mice.

To investigate the impact of daily rhVEGF-B treatment on cell survival, we examined the linear density of Purkinje cells at P30 (Figure 9). We identified a significant decrease in all PCD experimental groups (*p* PCD = 0.003; *p* PCD-NaCl < 0.001; *p* PCD-rhVEGFB = 0.002; *p* value relative to WT) in comparison with WT animals (Figure 9E). Similarly, to the analyses of rhIGF-1 treatment, we performed a new statistical comparison of the data considering only PCD groups, due to the large disparity of the data between them and WT animals. However, we did not detect any difference between PCD, PCD-NaCl, and PCD-rhVEGFB mice (Figure 9F; *p* > 0.05).

Henceforth, the daily administration of rhVEGF-B does not elicit any neuroprotective impact in PCD mice. This result agrees with previous behavioral data and led us to exclude this increased frequency of rhVEGF-B administration as a treatment of choice. In this sense, the success of the treatment with rhVEGF-B administered every other day suggests that excessive doses of rhVEGF-B have a detrimental effect.

Therefore, we highlight the administration of rhVEGF-B every other day as the treatment of choice to delay Purkinje cell death in the PCD mutant mouse.

## 3. Discussion

In this study, we assessed the potential therapeutic use of neurotrophic factors against severe, specific and fast neuronal death, like the cerebellar neurodegeneration in the PCD mouse. After an initial screening of the variations in these factors along such degenerative processes, we evaluated the neuroprotective properties of human recombinant proteins of IGF-1 and VEGF-B, which were found to be altered in this model. We have confirmed that the treatment with rhIGF-1 does not improve the degenerative condition of PCD mice. Conversely, the treatment with rhVEGF-B effectively delayed Purkinje cell degeneration and reversed behavioral impairments.

### 3.1. Searching for Neurotrophic Factors with Potential Therapeutic Use: Alterations in IGF-1 and VEGF-B Expression in the PCD Mutant Mouse

Currently, there is a considerable interest in exploring the role of neurotrophic factors as potential therapies to counteract neuronal death in neurodegenerative diseases [25] due to their effects on neuronal survival, axonal growth, synaptic plasticity, and neuronal function [24]. However, it is important to consider the timing and characteristics of each disease when using these treatments to optimize research and avoid the waste of resources in low-effective approaches. Therefore, first of all, to assess those putative neurotrophic factors that may be useful against neuronal death in our model, we carried out gene and protein analyses of BDNF, IGF1, VEGF-A, and VEGF-B along the period of cerebellar degeneration of PCD mice. Our analysis only revealed differences in the expression of IGF-1 and VEGF-B. IGF-1 changes were observed at P25 and P40, that is to say, at the end of the cerebellar neurodegenerative process. At P25, the cerebellum seems to require significant amounts of IGF-1, resulting in increased gene and protein expressions in this region, while protein levels in the blood decrease. It should be noted that although IGF-1 is produced in all cell types of the CNS [33,34], its mRNA expression in the brain is relatively low, so its peripheral production is crucial [35]. Indeed, this peripheral production supposes more than 70% of the total IGF-1 synthesis [36]. At this time in PCD mice, a very rapid and aggressive Purkinje cell death is occurring [7,12], and IGF-1 has been found to have neuroprotective properties that support axonal regeneration, remyelination, and other benefits [37,38]. Thus, one plausible hypothesis is that peripheral IGF-1 production is traveling to the brain to meet its high demands, considering that the brain’s intrinsic IGF-1 production levels are not enough for neuroprotection against a massive neuronal death [35]. However, at P40, we saw the opposite situation: despite the increase in gene expression in the cerebellum of PCD mice, the concentration of IGF-1 protein in this region remains similar to the WT one, while it increases in plasma. At this age, Purkinje cell degeneration can be considered complete, but the effect of the PCD mutation causes the loss of other cell populations at this age [7,8,10]. Thus, due to the systemic effects of this neurotrophic factor [39,40], this cerebellar genetic demand for IGF-1 may be due to other pathophysiological situations occurring outside the cerebellum and even outside the brain [4], the IGF-1 protein passing into the blood to satisfy those demands.

Fluctuations were found in VEGF-B at P15 and P20. The onset of cerebellar pre-neurodegeneration in the PCD mouse is estimated to be around P15, with cytoplasmic and nuclear changes in Purkinje cells [6,12,13,14]. Then, the increase in gene and protein VEGF-B expression at this age may be related to the onset of this stage of cerebellar degeneration, as it has been shown that VEGF-B increases after neuronal damage [41] and plays an important role in the process of cellular repair [42]. In addition, this fluctuation is likely to be mainly local, as VEGF-B production occurs in most neural cells [43,44].

On the other hand, at P20, when Purkinje cell death begins (the genuine degenerative stage), we found a reduction in *Vegf-B* gene expression in the cerebellum and in its corresponding protein levels in the blood plasma in mutant mice. Conversely, the protein levels in the cerebellum remained unchanged. In turn, the fact that VEGF-B is mainly produced in neuronal cells may explain the decrease in plasma levels following a decrease in gene expression. Given the crucial role of this neurotrophic factor in protecting the brain, a reduction in its production could have severe consequences in a neurodegenerative environment. Our hypothesis is that, despite the decline in *Vegf-B* gene expression, the system strives to maintain the protein in the cerebellum, where it is most needed, by reducing it at the peripheral level and potentially in other brain regions. Another function of VEGF-B is an anti-apoptotic effect, which promotes cell survival. This effect is achieved through its interaction with the VEGFR1 receptor, inducing the expression of anti-apoptotic genes such as Bcl-2, and inhibiting pro-apoptotic genes such as those of the BH3 subfamily and cell death-related proteins, including p53 and members of the caspase family [45]. Interestingly, a decrease in anti-apoptotic proteins such as BCL-2 can be observed at P22 in PCD mice [46]. In this sense, VEGF-B could create an anti-apoptotic and neuroprotective environment while its expression is upregulated, which is why, after the decrease in its expression, an increase in cell death appears in PCD mice, probably due to alteration raise in the levels of apoptotic proteins (apart from the accumulative damage that mutant Purkinje cells suffer).

In summary, each neurotrophic factor altered by the PCD mutation seems to be associated with a stage of cerebellar degeneration. First, we found changes in VEGF-B corresponding to the early stages of this process: P15, when pre-neurodegeneration starts, and P20, when neuronal death begins. Then, IGF-1 increases its expression at later stages: P25, the time in which the Purkinje cells are dying, and P40, the end of the degenerative process in the cerebellum.

Using these data, we designed an experimental approach adapted to the changes associated with the degenerative process of PCD mice. Moreover, before starting the treatments, an analysis of IGF-1 and VEGF-B receptors was conducted, confirming their presence in Purkinje cells, the targeted neuronal type. Previous studies have shown that the IGF1R is located in the cerebellum [35], but to our knowledge the expression of VEGFR1 in Purkinje cells has not been characterized. These analyses ensured that each treatment was specifically adapted to the cerebellar neurodegeneration of the PCD mutant mouse.

### 3.2. rhIGF-1 Treatment Does Not Provide a Neuroprotective Effect in PCD Mice

In recent years, IGF-1 has been studied in the context of neurodegenerative diseases due to its neuroprotective and neurogenic properties [37,47,48]. This study assessed the putative neuroprotective effect of rhIGF-1 on cerebellar neurodegeneration in the PCD mouse, and its effect on motor, cognitive and social behaviors. The cerebellum has traditionally been considered a structure of sensorimotor integration [49,50,51]. However, in recent decades, the cerebellum has been shown to be involved in other cognitive, affective and social functions, as well as in those related to language, attention and memory [6,52,53,54]. In this context, as described in other animal models, including the PCD mouse, cerebellar impairments can lead to alterations in all these functions [6,15,18,19,55,56]. Moreover, such alterations have been detected in humans with CONDCA, carriers of a mutation analogous to that of PCD mice [1,2,4]. Then, any neuroprotective effect on the Purkinje cell loss should have an impact on these different behaviors. Unfortunately, no improvements were found in any of the tests performed after rhIGF-1 administration. In all tests, the mutant mice treated with rhIGF-1 behaved similarly to untreated PCD mice. Moreover, after sacrificing the animals at P30, we also found no improvements in Purkinje cell survival, confirming that rhIGF-1 treatment does not have a neuroprotective effect on cerebellar degeneration in PCD mice and therefore does not restore altered cerebellar functions.

If we look at the existing literature, our data are quite surprising. Several studies have shown an improvement in motor function after IGF-1 treatment in animal models of ataxia with different etiologies [28,57,58,59,60,61,62,63,64]. More specifically, several authors reported an amelioration of ataxia in PCD mice after IGF-1 treatment, resulting in improved motor coordination [28,30], but it is unclear whether these effects occurred by rescuing degenerating neurons or by modulating the function of surviving neurons. Moreover, in 2003, Nahm et al. showed that exogenous IGF-1 administration did not prevent or rescue Purkinje cell death in their model of cerebellar neurodegeneration [29]. This suggests that the improvements in locomotion observed in previous studies following IGF-1 treatment may be due to an enhancement at the extracerebellar level or via the surviving Purkinje cells. In the case of extracerebellar improvement, we know that IGF-1 is taken up by neurons in the inferior olive and transported via climbing fibers to Purkinje cells [65,66]. Therefore, it is not only necessary to maintain optimal IGF-1 levels in the cerebellum, but it also seems to be important that IGF-1 is transported from the inferior olive to the Purkinje cells [61]. This fact supports the aforementioned explanation of the fluctuations in the IGF-1 levels inside and outside the cerebellum. In the case of the second option (i.e., restoring motor behavior by boosting surviving neurons), it appears that IGF-1 does not rescue dying Purkinje cells, but diverse findings suggest that this treatment is most effective when started in pre-neurodegenerative stages, exerting a protective rather than a rescue effect [29]. Perhaps a different, earlier dosing schedule would have yielded more satisfactory results. Unfortunately, the variation in IGF-1 levels throughout the cerebellar degenerative stage of the PCD mouse led us to exclude a priori this possibility, to adjust this research to our objectives and work plan.

In conclusion, although there is some evidence that IGF-1 has a neuroprotective effect on cerebellar degeneration, our data did not show significant improvements in our model and conditions. Therefore, we did not perform further analyses with this neurotrophic factor.

### 3.3. Treatment with rhVEGF-B Has a Neuroprotective Effect in PCD Mice

Unlike other members of the VEGF family, the VEGF-B lacks general angiogenic activity, and primarily exerts neuroprotective [41,67] and metabolic effects [68,69,70]. However, despite a growing number of studies demonstrating its neuroprotective properties, its impact on certain neurodegenerative diseases remains unclear. In this study, we have shown that the administration of rhVEGF-B improved motor, cognitive, and social skills altered in PCD mice. First, we observed a partial recovery of motor performance in mutant mice. These results are comparable to those observed in animal models of Parkinson’s disease, where preventive treatment with rhVEGF-B improved motor behavior [70,71]. Similarly, the treatment completely restored some of the functions altered in the PCD model, such as recognition memory and social preference. Both functions remained similar to those of WT mice until P30. Our data demonstrate for the first time the neuroprotective effect of rhVEGF-B treatment in cognitive and social behavior, as—to our knowledge—no studies have addressed this neuroprotective effect in models of cerebellar ataxia or other cognitive-affective disorders.

Furthermore, analysis of Purkinje cell survival at P25 demonstrated that the treated mice exhibited a comparable Purkinje cell density to WT mice. That is to say, the VEGF-B treatment virtually stopped the neuronal death for one entire week (from P18 to P25) in a model of disease in which the complete degeneration takes only four weeks longer [7]. However, at P30, although we found a significant increase in the number of Purkinje cells in treated mice compared to PCD mutants, the neuronal density was significantly lower than in WT mice. Therefore, the initial protection was not maintained indefinitely over time. These data are consistent with what has been observed in animal models of Parkinson’s disease, where they describe an increase in neuronal density just after treatment, accompanied by an improvement in behavior [41,70,71]. As far as the time course of the effects of rhVEGF-B is concerned, our data suggest that the neuroprotective effect occurs during treatment. These effects seem to last a little longer, since at P25, although the treatment had ended, we still found a clear neuroprotective effect. On the contrary, at P30, ten days after the end of the treatment and due to the rapid and aggressive degeneration associated with the PCD mutation [7,12,72], Purkinje cell death is again evident, although in an attenuated or delayed manner. In this regard, we know that the protective effects of VEGF-B are preventive. Studies in animal models of Parkinson’s disease have shown that administration of VEGF-B before the onset of damage leads to an improvement in both behavior and cell density. However, administration after the onset of degeneration, is not associated with cellular protection [70]. We observed an improvement in PCD mice when treatment was started at P10, before the onset of neuronal death. However, this is not maintained over time, so it would be interesting to evaluate longer or repeated treatments to understand their effects on the cell death process and achieve a higher level of neuroprotection.

Although the mechanism by which VEGF-B exerts its neuroprotective effect is not known, it appears to be related to the activation of the VEGFR1 receptor, which is expressed in neurons [67,69], especially in Purkinje cells, as we have verified. Given the relationship between VEGF-B and the suppression of cell death through this receptor [45], we decided to perform an apoptosis study in the Purkinje cell layer. At P25, we found an increase in the density of apoptotic cells labeled with TUNEL in both treated and untreated PCD groups. This increase was greater in the untreated PCD mice, while the treated mice were midway between the WT and PCD groups. On the other hand, at P30, we saw that apoptotic cells increased significantly in the mutant mice treated with rhVEGF-B, reaching higher levels than in WT and PCD mice. Therefore, these data suggest that the neuroprotective effect of VEGF-B is mediated by a delay of apoptosis [45], at least until P25. The increase in apoptosis observed at P30 in the treated mice could be because we found a higher cell density—thanks to the neuroprotection achieved—that are susceptible to DNA damage and continue their degenerative process. Thus, the treatment may slow down the rate of neuronal death, allowing a greater number of apoptotic cells to be observed at an advanced age of neurodegeneration. In this sense, similar characteristics were observed in PCD mice treated with the endocannabinoid oleylethanolamide, where the mutant mice displayed partial neuroprotection at P30 [20]. In conclusion, the administration of rhVEGF-B completely delayed Purkinje cell death at P25, their effects lasting up to P30, leading to a partial improvement in motor coordination and a complete improvement in cognitive and social functions.

### 3.4. Overdosage of rhVEGF-B Has Adverse Effects in the PCD Mutant Mouse

Once the neuroprotective effects of rhVEGF-B on cerebellar neurodegeneration were confirmed when administered every two days, and since there is not enough literature on the optimal schedule of administration of this neurotrophic factor, we decided to increase its frequency of administration. Then, to improve the effects of rhVEGF-B, we treated a new group of animals with a daily frequency at the same ages described, from P10 to P20. This daily administration was similar to the schedule used with other neurotrophic factors [73,74,75].

In this case, we found no improvement in any of the variables analyzed. Firstly, in motor, cognitive, or social behavioral tests, daily treated mutants showed similar results to untreated PCD mice. We also found no effect in cell survival analysis, but contrary to expectations, cell density in mice treated daily with rhVEGF-B was like untreated PCD mice. Occasionally, a high frequency of a particular treatment may be not tolerated, instead being better tolerated at lower frequencies or doses [76]. This means that the beneficial effect of rhVEGF-B seems to follow the typical inverted U-shaped curve with respect to the dose administered. Other pharmacological treatments in the PCD mouse have demonstrated comparable patterns. This is the case of oleylethanolamide, whose chronic treatment did not induce improvements in Purkinje cell density in PCD mice, while more spaced out or even single doses were much more effective [20]. In the case of neurotrophic factors, an excessive dose can sometimes have a detrimental effect on cell development and survival [77], which can be a possible explanation for our results. Therefore, it is essential to carefully regulate the administration of this type of substance. Based on the results obtained, we verified that doubling the rate of administration of rhVEGF-B did not improve its effect on neurodegeneration in the PCD mouse.

In conclusion, although the precise molecular mechanisms through which rhVEGF-B exerts its effects require further study, its neuroprotective nature is evident at both histological and behavioral levels. This neuroprotective effect appears to be due to a reduction in apoptosis in Purkinje cells. These results provide evidence for the use of rhVEGF-B as a potential pharmacological molecule to reduce neuronal death associated with aggressive and fast neurodegenerative diseases. In particular, further studies on this topic may be useful to address if children affected by CONDCA may benefit from the administration of this substance, especially in the early stages of the illness.

## 4. Materials and Methods

### 4.1. Animals

C57/DBA mice obtained from Jackson ImmunoResearch Laboratories were housed at the animal facilities of the University of Salamanca at a constant temperature and humidity, with a 12/12 h light/dark photoperiod. They were fed with water and special rodent chow ad libitum (Rodent toxicology diet, B&K Universal G.J., S.L. Molins de Rei, Barcelona, Spain). As PCD mice are not suitable for breeding, the colony was maintained by mating heterozygous animals and genotyping their offspring as previously described [23].

All animals were housed, manipulated, and sacrificed in accordance with current European (Directive 2010/63/EU, Recommendation 2007/526/CE) and Spanish Legislation (RD118/2021, Law 32/2007); the experiments were approved by the Bioethics Committee of the University of Salamanca (Reference #613). All efforts were made to minimize animal suffering and to use the fewest animals required to produce statistically relevant results.

### 4.2. Experimental Procedure

We first provide a summary of the general procedure to facilitate the comprehension of the experiments performed in this study.

Initially, a search for neurotrophic factors with putative therapeutic effect was carried out, encompassing the analysis of the expression of four of these molecules in the cerebellum and plasma of both WT and PCD mice. The findings of this initial study contributed to the creation of an experimental design in which we administered the recombinant human (rh) proteins of those neurotrophic factors altered in the PCD mutant mouse: IGF-1 and VEGF-B.

The impact of both treatments was assessed by behavioral tests and by estimating the survival of Purkinje cells (Appendix A). Due to the positive effects of administering rhVEGF-B every other day, further investigation into additional aspects of its neuroprotective influence was carried out: analysis of apoptosis and the condition of the skeletal muscle, considering the enhancements in motor behavior (Appendix A).

### 4.3. Molecular Analyses

To find out whether the PCD mutation causes changes in the expression levels of neurotrophic factors, changes in gene and protein expressions of BDNF, VEGF-A, IGF-1, and VEGF-B were analyzed.

Animals were sacrificed by cervical dislocation and decapitation (n = 6 animals per experimental group and age). The cerebellum was sectioned along the sagittal midline to obtain separately the two cerebellar hemispheres with a half of the vermis. Then, these two parts were snap frozen in liquid nitrogen and stored at −80 °C. The left hemisphere/vermis was used for gene analysis by real time quantitative polymerase chain reaction (RT-qPCR), while the right hemisphere/vermis was used for protein analysis by Enzyme-Linked ImmunoSorbent Assay (ELISA).

#### 4.3.1. Gene Analyses

These experiments were performed in both WT and PCD mice throughout the neurodegenerative process, from P10, when any degeneration is not observed in PCD animals, to P40, when neurodegeneration is considered complete. In between, P15, P20, P25, and P30 ages were also analyzed.

RNA was isolated from the left hemicerebellum using the PureLinkTM RNA Mini Kit (Invitrogen, Waltham, MA, USA) and the PureLinkTM DNase Set (Invitrogen). 1000 ng of RNA was converted to cDNA using the High-Capacity cDNA Reverse Transcription Kit (Applied Biosystems, Foster City, CA, USA). RT-qPCR was performed using PowerUp SYBR Green Master Mix (Applied Biosystems) and the specific pairs of primers for the neurotrophic factors listed in Table 1. RT-qPCR assays were carried out in accordance with the manufacturer’s instructions on the QuantStudioTM 7 Flex RT PCR System (Applied Biosystems), utilizing the indicated amplification cycling conditions. Triplicate analyses were performed on each biological sample. Glyceraldehyde-3-phosphate dehydrogenase (Gapdh) expression was employed as the endogenous control to normalize the data, and the fold change expression was calculated based on the normalized Cycles to Threshold (Ct) values.

#### 4.3.2. Protein Analyses

Protein analysis was carried out in a complementary manner and as a confirmation of the data obtained in the gene study by RT-qPCR, since it provides a quantitative measure of the protein production of the genes of interest. For this reason, we only conducted this analysis at the crucial ages of neurodegeneration, specifically P15 (pre-neurodegeneration) and P25 (peak of degeneration), and at those time points when changes in gene expression were observed by RT-qPCR (P40 for *Igf-1* and P20 for *Vegf-B*). Additionally, protein expression was examined in blood plasma at the same aforementioned ages. To achieve this, the right hemicerebellum of the same mice used for gene analysis by RT-qPCR were processed. The plasma was collected using ethylenediaminetetraacetic acid (EDTA) as anticoagulant, centrifuged for 10 min at 5G and finally, the supernatant was removed immediately and assayed or stored at −80 °C.

First, total protein concentration of each sample was determined in triplicate by the method of Bradford. Then, the levels of BDNF, VEGF-A, IGF-1, and VEGF-B in tissue homogenates and plasma samples were determined by double-antibody sandwich ELISA according to the manufacturer instructions (IGF-1, MG100 and VEGF-A, MMV00, R&D Systems, Minneapolis, MN, USA; BDNF, EM0020, FineTest, Wuhan, China and VEGF-B, SEA144Mu, USCN, Wuhan, China). The protein levels were expressed in pg of the specific neurotrophic factor/mL of total protein.

### 4.4. Drug Administration

For pharmacological treatment we used rhIGF-1 (Cat: 100-11, Peprotech, Cranbury, NJ, USA) and rhVEGF-B (Cat: 100-20B, Peprotech). Both rhIGF-1 and rhVEGF-B were freshly prepared to avoid degradation of the compounds. Each drug was diluted in NaCl 0.9% (*w*/*v*) to the concentration of use, which was 25 µg/mL for rhIGF-1 and 1.5 µg/mL for rhVEGF-B. Both substances were administered intraperitoneally in a volume of 10 µL/g animal weight. For rhIGF-1 treatment, a daily administration was carried out from P20 to P30, while for rhVEGF-B injections were performed every other day from P10 to P20. In addition, one group of animals for each treatment was administered with 0.9% (*w*/*v*) NaCl only to exclude possible effects due to administration procedures. Finally, based on the results obtained after the administration of rhVEGF-B every other day, a supplementary treatment involving daily administration of this neurotrophic factor from P10 to P20 was also implemented. All treated animals were dosed in the morning between 9 and 12 AM and their body weight was monitored throughout the experimental procedures.

### 4.5. Behavioral Analyses

A battery of behavioral tests related to motor, cognitive and social functions were performed at different time points throughout the degenerative process of PCD mice (n = 8 animals per experimental group and age). The ages depended on the study performed (indicated in each section of Results).

The Novel Object Recognition test was performed to evaluate recognition memory in rodents [6,20,78,79]. On the first day of testing, two identical objects were placed in opposite corners of the cage and presented to each animal for 10 min (the familiar object). Then, one of the familiar objects was replaced with a novel object and the test began. We analyzed the percentage of time spent exploring the familiar and the novel object, and the discrimination index (DI). Since mice are exploratory animals, in normal conditions they tend to spend more time investigating novel objects [6,20,79]. In turn, the DI represents the ability to discriminate between the two objects, and it can range from +1 to −1 depending on the exploration time. A positive DI (>0) indicates that the animal spent more time exploring the new object, thereby discerning both objects. Alternatively, a neutral or negative DI (≤0) denotes an absence of discrimination, resulting in compromised recognition memory. DI was calculated as previously described in Ref. [20].

The three-chamber social preference test was performed to analyze social behavior in a white Plexiglas box divided into three connected rooms [6,20,80]. After the habituation process, the analyzed mouse was placed in the central room, while the adjacent rooms contained either a mouse or an object, both covered with a container that allowed social and olfactory interaction, but not direct physical interaction. The percentage of time spent interacting with the animal or the object was calculated. In the absence of any disorder, mice, as social animals, tend to spend more time with their peers than with an inanimate object [6,20]. We also calculated the sociability index (SI), which provides information about the social preference of the mice analyzed. The SI score varies between +1 and −1, whereby positive values (SI > 0) signal a preference for social interaction, values close to 0 suggest no preference, and negative values (SI < 0) indicate social avoidance tendencies [20].

The Rotarod test (LE8200, Panlab, Barcelona, Spain) was used to analyze motor skills and coordination, as previously described [6,81]. This test consists of a rod rotating at a speed of 0.06 rpm/s, with accelerations ranging from 4 to 40 rpm. We specifically utilized the latency to fall parameter, which refers to the duration of the animal on the rod prior to falling [6,20]. This parameter was analyzed as an average of 7 trials with 20 min rest between trials for each mouse and day of testing.

### 4.6. Histological and Cytological Analyses

#### 4.6.1. Tissue Extraction and Processing

Animals were deeply anesthetized. Then, they were injected into their left ventricle with 100 µL of heparin (1000 U/mL), and perfused intracardially with 0.9% NaCl (*w*/*v*), followed by modified Somogyi’s fixative (0.4% *w*/*v* paraformaldehyde and 15% *v*/*v* saturated picric acid in 0.1 M phosphate buffer, PB) for 15 min. The encephalon and a muscle (quadriceps femoris) were extracted and washed with PB and cryoprotected with 30% sucrose (*w*/*v*) in PB. The cerebellum was dissected and sectioned using a freezing-sliding microtome (Jung SM 2000, Leica Instruments, Wetzlar, Germany) obtaining 30 µm-thick parasagittal sections, which were stored in cryoprotective solution at −20 °C until their use.

Muscle was sectioned using a Microm HM 560 cryostat (Thermo Scientific, Waltham, MA, USA) to obtain 8 µm-thick sections oriented either longitudinally or transversely to the major axis of the muscle, and they were stored at −20 °C until they were used for immunostaining.

For immunofluorescence analyses, four equidistant parasagittal sections of the cerebellar vermis were washed in phosphate-buffered saline (PBS; 3 × 10 min) and incubated for 72 h at 4 °C with continuous rotation in a medium containing 0.2% (*v*/*v*) Triton X-100 (Probus S.A., Geneva, Switzerland), 5% (*v*/*v*) normal donkey serum (Sigma-Aldrich, St. Louis, MO, USA), and the following primary antibodies diluted in PBS: mouse anti-calbindin D-28k (Cb28k; 1:1000; Swant, Bellinzona, Switzerland), rabbit anti-IGF1R (1:1000; Abcam), rabbit anti-VEGFR1 (1:1000; Abcam, Cambridge, MA, USA), and rabbit anti-caspase 3 (1:250; Sigma-Aldrich). Then, sections were incubated with a Cy2 and Cy3-conjugated secondary antibody (1:500; Jackson ImmunoResearch, West Grove, PA, USA) in PBS for 2 h at room temperature and counterstained with 4′,6-diamidino-2-phenylindole (DAPI; 1:10,000; Sigma-Aldrich) to identify cell nuclei.

For TUNEL technique, sections were washed with PBS (3 × 10 min) and fixed with ethanol:acetic acid 2:1 (*v*/*v*) for 5 min. Then, they were permeabilized with 0.2% (*v*/*v*) Triton X-100 and 0.1% (*w*/*v*) sodium citrate in distilled water for 15 min. After three washes with PBS (10 min each), sections were incubated in TUNEL buffer for 30 min. Terminal transferase (2 µL/mL; Roche, Basel, Switzerland) and biotinylated dUTP (1 µL/mL; Roche) were then added to the TUNEL buffer for 1.5 h at 37 °C. The reaction was stopped with sodium citrate buffer (2 × 10 min). Sections were washed in PBS (3 × 10 min) and incubated with Cy2-conjugated streptavidin (1:200; Jackson) diluted in PBS, and nuclei were counterstained with DAPI.

For skeletal muscle analysis, we used standard hematoxylin and eosin staining. First, the sections were washed in distilled water, they were stained with hematoxylin use solution (Panreac, Barcelona, Spain) for 5–10 min and washed again using distilled water. Eosin acid dye (Panreac) was then added for 2 min, and the tissue was dehydrated through increasing concentrations of ethanol (from 50% to 100% *v*/*v*) and xylene. Finally, the sections were mounted with Entellan (Merck, Darmstadt, Germany) and a coverslip for further analysis.

#### 4.6.2. Microscopy Visualization and Quantifications

To study the effect of the different pharmacological treatments on neuronal survival and apoptosis, sections were observed using an Olympus Provis AX70 epifluorescence microscope coupled to an Olympus DP70 digital camera (12.5 MP, Olympus, Tokyo, Japan) and calbindin- or TUNEL-positive Purkinje cells were counted manually using 20× and 40× objectives, respectively. Only Purkinje cells with a clear Cb28k+ soma and dendritic arbor, and TUNEL-positively stained cells located in the Purkinje cell layer were included in the analyses. For both analyses, the length of the Purkinje cell layer was measured by using the Neurolucida software version 8.0 (MBF Bioscience, Williston, VT, USA), and the linear density was subsequently calculated. These assessments were conducted at P25 and/or P30 (n = 6 animals per experimental group and age), as detailed in the Section 2. In general, all cell counts were performed considering the entire cerebellar vermis (lobes I to X).

Muscle analysis was performed both longitudinally and transversely using the quadriceps femoris (n = 6 animals per experimental group). For the tissular assessment, we acquired photographs of the muscle fibers of each section at 20× and then analyzed them using the ImageJ software version 1.54f for Windows (NIH, Bethesda, MD, USA). To do this, after calibrating the image, we first separated the colors using the color deconvolution option. Then, we analyzed and created the different regions of interest (ROI) by segmenting the elements according to the size of the fiber. In the longitudinal analysis, the program converted each muscle fiber into an ellipse and gave us information about its major and minor axes, i.e., the estimated length and width of each muscle fiber. For the muscle transverse-sectional analysis, we analyzed the mean cross-sectional area of each fiber using a similar computer analysis.

### 4.7. Statistical Analyses

Considering the size of the samples, we have generally performed non-parametric tests, using the Mann–Whitney or Kruskal–Wallis U tests, depending on the number of experimental groups. However, for the statistical analysis of the Rotarod test, the one-way repeated measures ANOVA test was used to determine the interaction between the variables analyzed (time and experimental group), since learning may influence the performance through the test.

In all cases, the minimum level of statistical significance was set at *p* < 0.05. All statistical analyses were performed with the statistical program SPSS version 28 for Windows (IBM, Armonk, NY, USA).

## Figures and Tables

**Figure 1 ijms-26-00538-f001:**
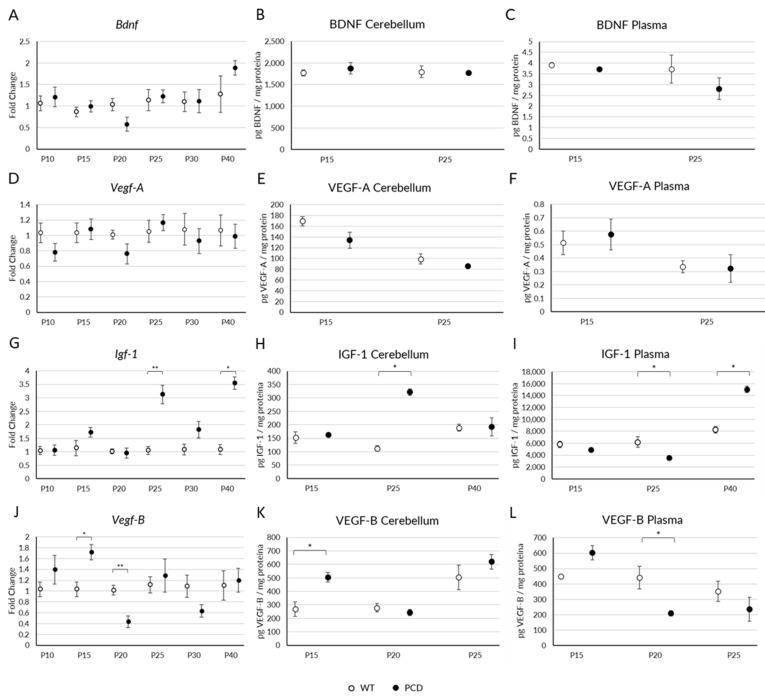
Analyses of gene and protein expression of BDNF, VEGF-A, IGF-1, and VEGF-B in WT and PCD mice. (**A**–**C**) Graphical representation of gene (**A**), protein expression in cerebellum (**B**), and blood plasma (**C**) of BDNF. (**D**–**F**) Analysis of gene (**D**), protein expression in cerebellum (**E**), and blood plasma (**F**) of VEGF-A. (**G**–**I**) Graphical representation of gene (**G**), protein expression in cerebellum (**H**), and in blood plasma (**I**) of IGF-1. (**J**–**L**) Analysis of gene (**J**), protein expression in cerebellum (**K**), and blood plasma (**L**) of VEGF-B. Note that at P25 there is an increase in both gene and protein expression of IGF-1 in the cerebellum of PCD mice; however, the protein levels decrease in the blood plasma. At P40, there is an increase in gene but not in protein expression of IGF-1 in the cerebellum; conversely, a protein increase in blood plasma is also evident. For VEGF-B, we observed an increase in both gene and protein expression in the cerebellum at P15. Conversely, at P20, there is a decrease in gene expression in the cerebellum, accompanied by a reduction in protein levels in blood plasma in mutant mice. BDNF and VEGF-A did not presented variations in PCD mice. * *p* < 0.05; ** *p* < 0.01.

**Figure 2 ijms-26-00538-f002:**
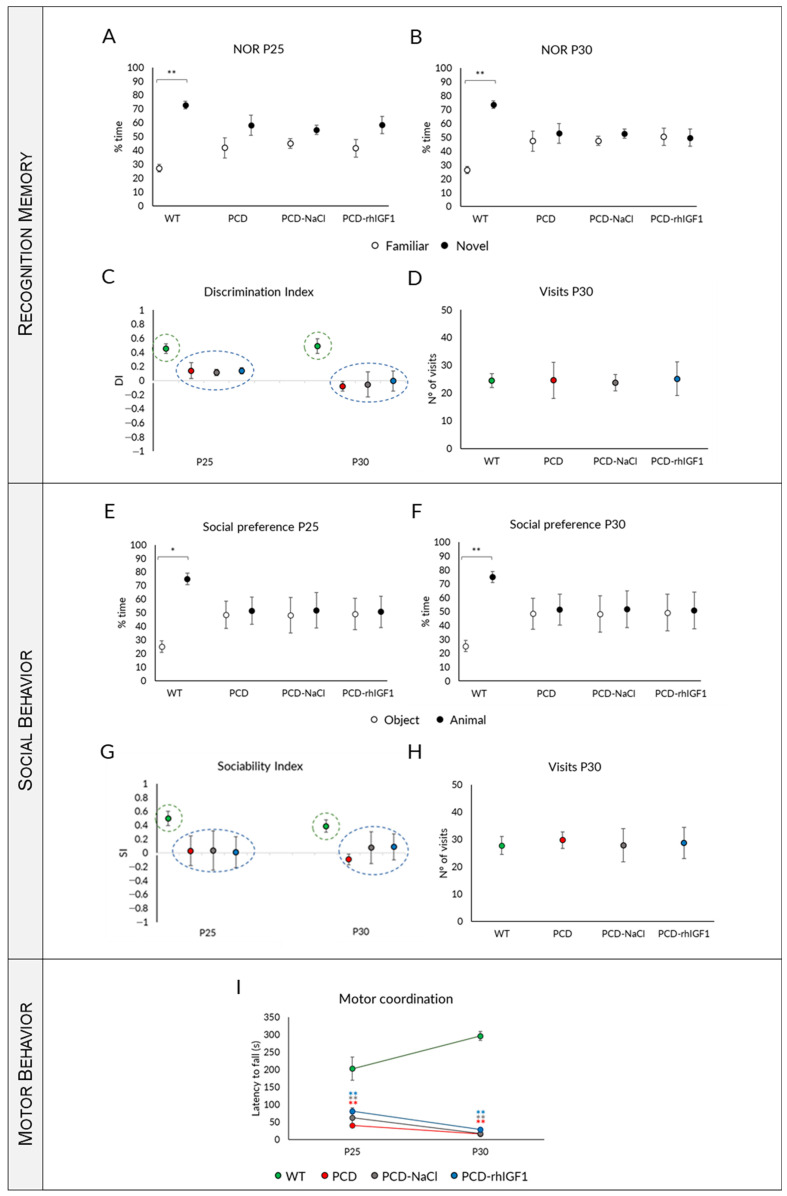
Effect of rhIGF-1 treatment on behavior throughout the neurodegenerative process in PCD mice**.** (**A**–**D**) Analysis of recognition memory: quantification of the percentage of time spent exploring the familiar or novel object at P25 (**A**) and P30 (**B**), discrimination index (**C**), and number of visits to both objects (**D**) of WT, PCD, PCD-NaCl, and PCD-rhIGF1 mice. (**E**–**H**) Analysis of social behavior: quantification of the percentage of time spent exploring the animal or the object at P25 (**E**) and P30 (**F**), sociability index (**G**), and number of visits (**H**) of WT, PCD, PCD-NaCl, and PCD-rhVEGFB mice. (**I**) Analysis of motor behavior at P25 and P30 of the abovementioned experimental groups. As it can be seen, rhIGF-1 treatment does not restore the altered functions in PCD mice. * *p* < 0.05; ** *p* < 0.01 for differences between WT and PCD experimental groups. The colors represent different PCD experimental groups.

**Figure 3 ijms-26-00538-f003:**
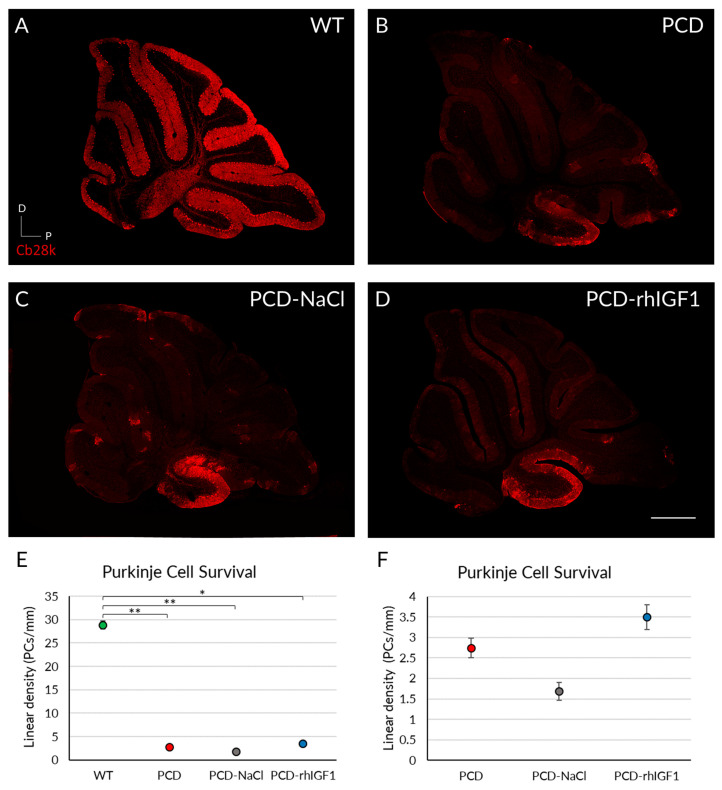
Effect of rhIGF-1 treatment at the histological level at P30. (**A**–**D**) Calbindin (Cb28k, in red) immunofluorescence of sagittal sections of the cerebellar vermis of WT (**A**), PCD (**B**), PCD-NaCl (**C**), and PCD-rhIGF1 (**D**) mice. (**E**) Graphical representation of Purkinje cell survival of the aforementioned experimental groups at P30. (**F**) Graphical representation of Purkinje cell survival of only PCD groups at P30. All mutant animals showed a significantly lower density of Purkinje cells compared to WT mice. Quantitative analysis corroborated such differences (**E**), and no effect of rhIGF1 was detected, even comparing only mutant animals (**F**). * *p* < 0.05; ** *p* < 0.01. Scale bar 500 µm.

**Figure 4 ijms-26-00538-f004:**
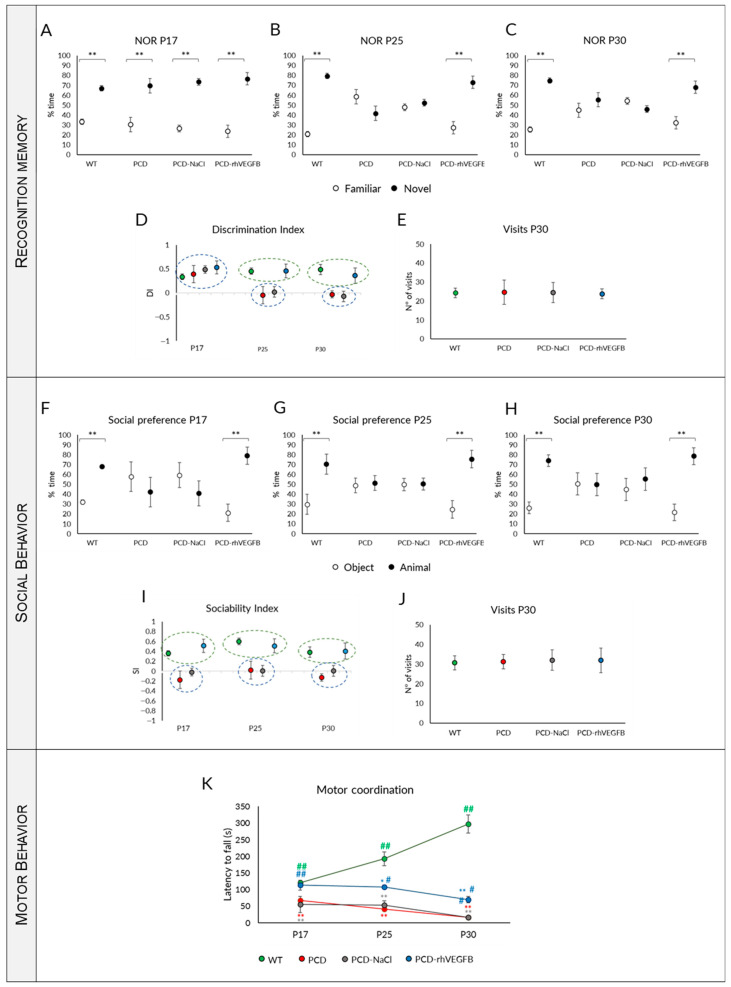
Effect of rhVEGF-B treatment on behavior throughout the degenerative process in PCD mice. (**A**–**E**) Analysis of recognition memory: Quantification of the percentage of time spent exploring the familiar and the novel object at P17 (**A**), P25 (**B**) and P30 (**C**), discrimination index (**D**), and number of visits to both objects at P30 (**E**) of WT, PCD, PCD-NaCl and PCD-rhVEGF-B mice. (**F**–**J**) Analysis of social behavior: Quantification of the percentage of time spent exploring the animal or the object at P17 (**F**), P25 (**G**), and P30 (**H**), sociability index (**I**) and number of visits to either the animal or the object at P30 (**J**) in the same experimental groups as above. (**K**) Analysis of motor behavior at P17, P25, and P30 of the same experimental groups aforementioned. Note that the treatment with rhVEGF-B completely restored recognition memory and social behavior, and partially improved motor behavior in PCD animals. * *p* < 0.05; ** *p* < 0.01, for differences between PCD and WT groups; # *p* < 0.05; ## *p* < 0.01 for differences amongst PCD groups. Each color represents a different experimental group.

**Figure 5 ijms-26-00538-f005:**
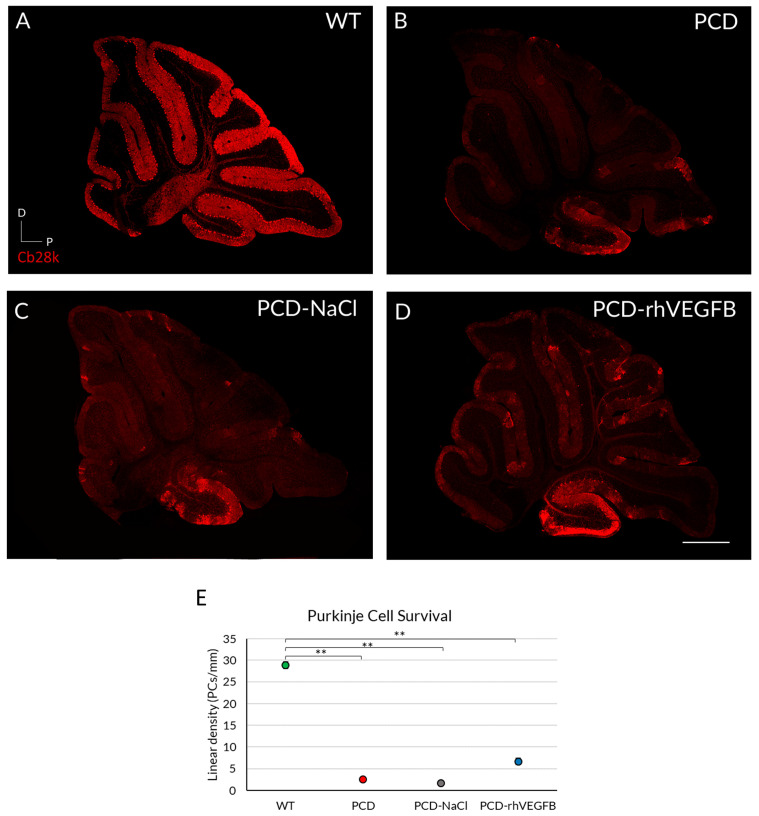
Effect of rhVEGF-B on Purkinje cell survival at P30. (**A**–**D**) Calbindin immunofluorescence (Cb28k, in red) of sagittal sections of cerebellar vermis from WT (**A**), PCD (**B**), PCD-NaCl (**C**), and PCD-rhVEGFB (**D**) mice at P30. (**E**) Quantification of cell survival (linear density) at P30 in experimental aforementioned groups. Note the higher number of neurons in PCD-rhVEGFB mice compared to the other PCD groups. ** *p* < 0.01. Scale bar 500 µm.

**Figure 6 ijms-26-00538-f006:**
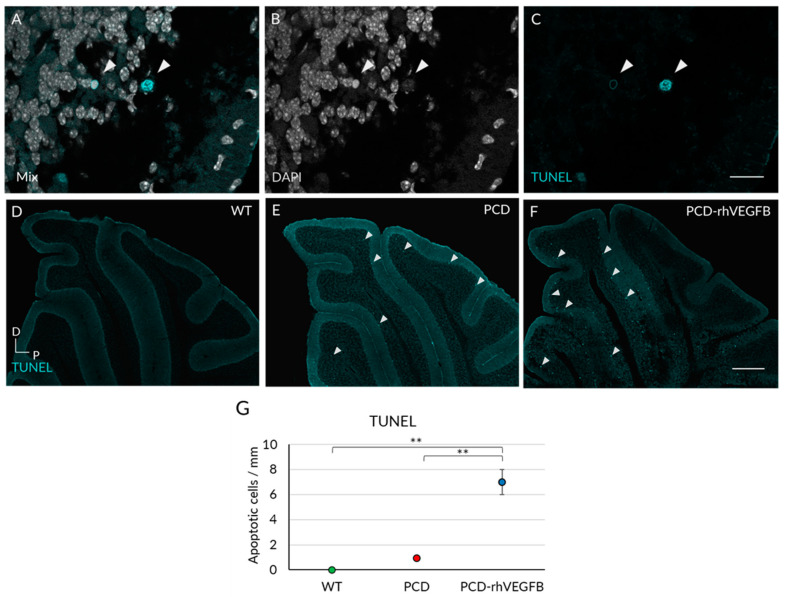
Effect of rhVEGF-B treatment on programmed cell death at P30. (**A**–**C**) Typical morphology of TUNEL-positive cells. (**D**–**F**) TUNEL labeling in sagittal slices of cerebellum from WT (**D**), PCD (**E**) and PCD-rhVEGFB (**F**) mice. (**G**) Quantification of apoptotic cell density in the Purkinje cell layer of WT, PCD and PCD-rhVEGFB mice. Note the increase in apoptotic cell density in PCD-rhVEGFB mice, which is higher than that observed in WT and PCD mice. The arrows indicate TUNEL-positive cells. ** *p* < 0.01. Scale bars 15 µm (**A**–**C**), 300 µm (**D**–**F**).

**Figure 7 ijms-26-00538-f007:**
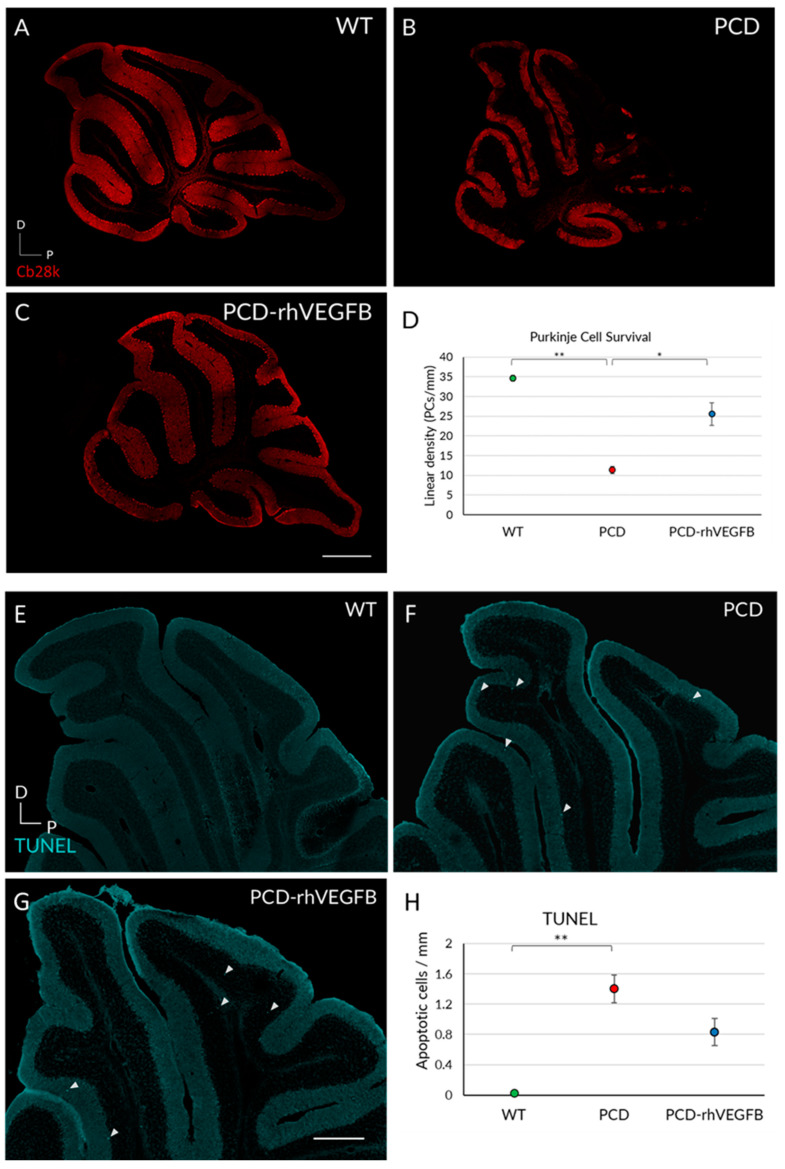
Effect of rhVEGF-B treatment on Purkinje cell survival and programmed cell death at P25. (**A**–**D**) Calbindin immunofluorescence (Cb28k, in red) of sagittal sections of cerebellar vermis from WT (**A**), PCD (**B**), and PCD-rhVEGFB (**C**) mice at P25. (**D**) Quantification of cell survival (linear density) at P25 in WT, PCD, and PCD-rhVEGFB. PCD-rhVEGFB mice presented a higher density of neurons compared to the other PCD groups, and without showing differences with WT animals. Scale bar 500 µm. (**E**–**H**) TUNEL labeling of sagittal sections of the cerebellum in WT (**E**), PCD (**F**), and PCD-rhVEGFB (**G**) mice. (**H**) Quantification of apoptotic cell density in the experimental groups mentioned above. Note the effect of rhVEGF-B in reducing the apoptotic cell density of PCD-rhVEGFB mice to an intermediate level between untreated PCD and WT mice. The arrows indicate TUNEL-positive cells * *p* < 0.05; ** *p* < 0.01. Scale bar 300 µm.

**Figure 8 ijms-26-00538-f008:**
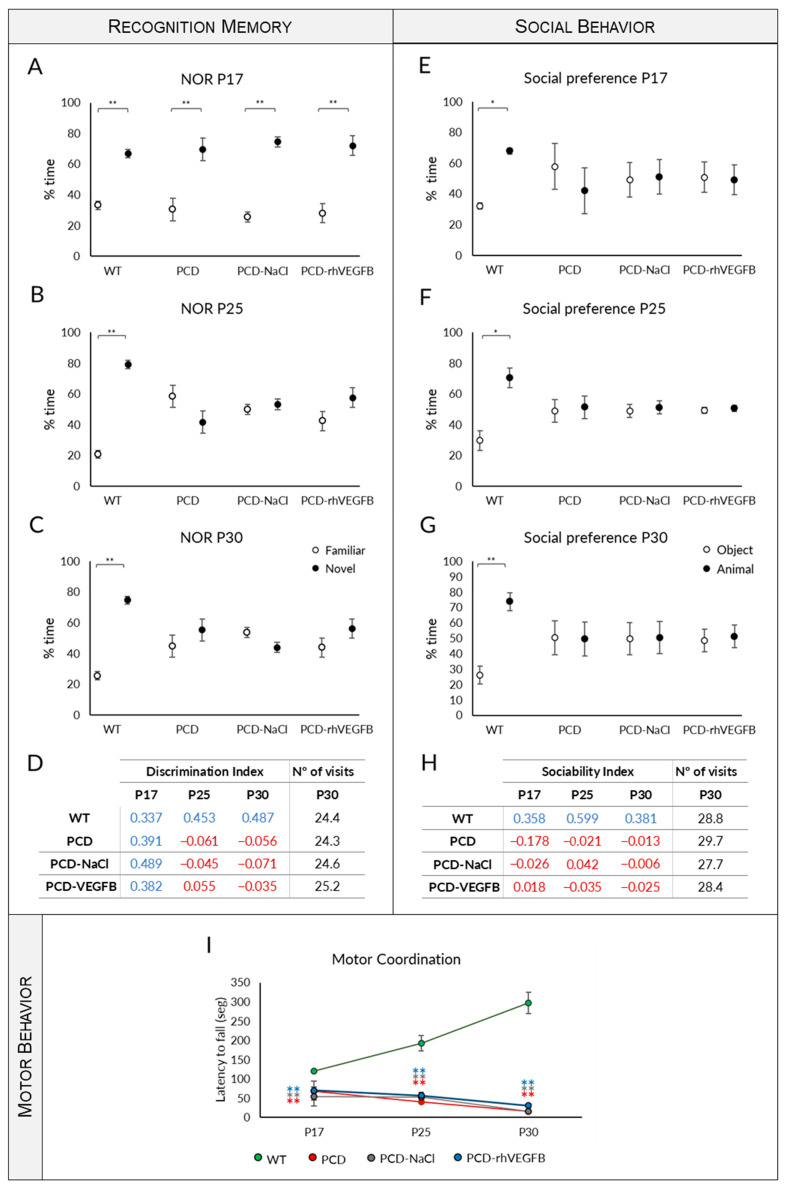
Effect of daily rhVEGF-B treatment on behavior throughout the neurodegenerative process in the PCD mouse. (**A**–**D**) Recognition memory analysis: (**A**–**C**) Quantification of the percentage of time spent exploring the familiar and novel object at P17 (**A**), P25 (**B**), and P30 (**C**) of WT, PCD, PCD-NaCl, and PCD-rhVEGF-B mice. (**D**) Summary table of the discrimination index at P17, P25, and P30 and the total number of visits to both objects at P30 in the aforementioned experimental groups. (**E**–**H**) Social behavior analysis: (**E**–**G**) Quantification of the percentage of time spent exploring the animal or object at P17 (**E**), P25 (**F**), and P30 (**G**) of WT, PCD, PCD-NaCl, and PCD-rhVEGFB mice. (**H**) Summary table of the sociability index at P17, P25, and P30 and the number of total visits to the animal and object at P30 in the different animal groups used. Blue color indicates positive values, while red color indicates negative or close to 0 values. (**I**) Analysis of motor behavior at P17, P25, and P30 of the abovementioned experimental groups. Note that daily treatment with rhVEGF-B did not improve any of the abilities analyzed, neither recognition memory, sociability, nor motor coordination. * *p* < 0.05; ** *p* < 0.01. Each color represents a different PCD experimental group.

**Figure 9 ijms-26-00538-f009:**
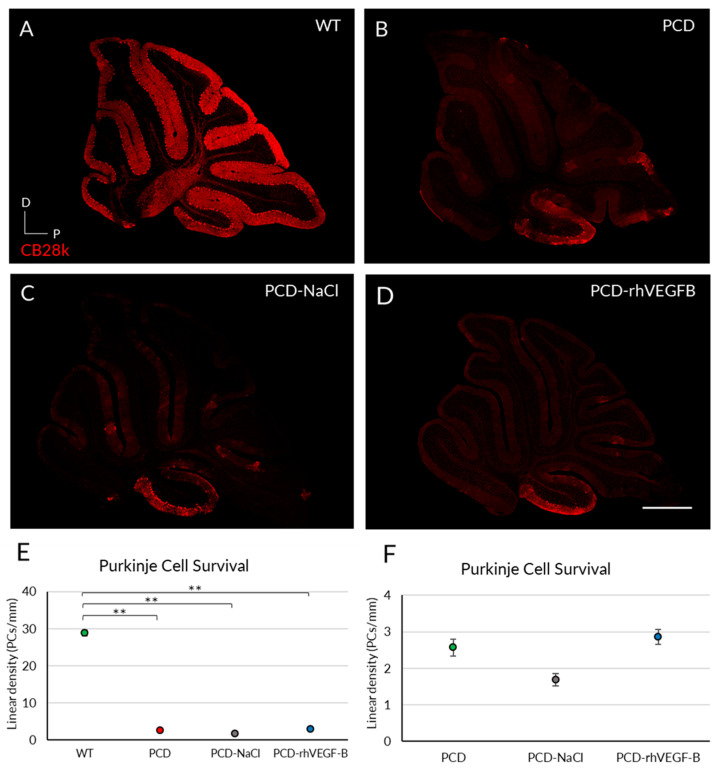
Effect of daily treatment with rhVEGF-B at a histological level at P30. (**A**–**D**) Immunofluorescence for calbindin (Cb28k, red) in sagittal sections of cerebellar vermis from WT (**A**), PCD (**B**), PCD-NaCl (**C**), and PCD-rhVEGF-B (**D**) mice. (**E**,**F**) Quantification of Purkinje cell survival (linear density) in the whole vermis of the four experimental groups analyzed (**E**) and excluding WT mice data (**F**). As can be seen, daily treatment with rhVEGF-B did not improve Purkinje cell survival, even analyzing PCD groups separately. ** *p* < 0.01. Scale bar 500 µm.

**Table 1 ijms-26-00538-t001:** List of RT-qPCR primer sequence details.

Gen Name	Forward Primer (3′-5′)	Reverse Primer (5′-3′)
*I* *gf-1*	GAAGACGACATGATGTGTATCTTTATC	AGCAGCCTTCCAACTCAATTAT
*B* *dnf*	GTGGTGTAAGCCGCAAAGA	AACCATAGTAAGGAAAAGGATGGTC
*V* *egf-A*	AATGCTTTCTCCGCTCTGAA	AAAAACGAAAGCGCAAGAAA
*V* *egf-B*	AGGAGGTTCGCCTGTGCT	GCTCAACCCAGACACCTGTAG
*G* *apdh*	GCCTATGTGGCCTCCAAGGA	GTGTTGGGTGCCCCTAGTTG

Abbreviations: *Igf-1*, insulin-like growth factor 1; *Bdnf*, brain-derived neurotrophic factor; *Vegf-A*, Vascular Endothelial Growth Factor A; *Vegf-B*, Vascular Endothelial Growth Factor B; *Gapdh*, glyceraldehyde-3-phosphate dehydrogenase.

## Data Availability

Data supporting the findings and conclusions of this study are available from the corresponding authors upon request.

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
