# Peer review of "Neuroprotective Effects of VEGF-B in a Murine Model of Aggressive Neuronal Loss with Childhood Onset"

_ijms, 2025, doi:10.3390/ijms26020538_

Round 1

Reviewer 1 Report

Comments and Suggestions for Authors

The manuscript titled NEUROPROTECTIVE EFFECTS OF VEGF-B IN A MURINE MODEL OF AGGRESSIVE NEURONAL LOSS WITH CHILDHOOD ONSET by Laura Perez-Revuelta proposes rhVEGF-B as a pharmacological agent to mitigate severe cerebellar neurodegenerative processes. The manuscript is highly interesting and novel; however, some parts require minor revisions:

1, Given that the VEGFB gene and protein are downregulated in PCD mice but increased in plasma, what is the underlying reason? Please discuss the potential mechanisms.

2, Since rhVEGF-B does not enhance Purkinje cell survival but improves motor function in mice, it is important to assess Purkinje cell morphology, as reported in the following study (PMID: 37082980).

3, rhVEGF-B reduces Purkinje cell apoptosis levels. What is the potential mechanism? I recommend the authors stain for Caspase-3 or other apoptosis-related markers following treatment to provide more insights.

Author Response

1, Given that the VEGFB gene and protein are downregulated in PCD mice but increased in plasma, what is the underlying reason? Please discuss the potential mechanisms.

The gene and protein analysis of VEGF-B revealed alterations at P15 and P20. At P15, gene expression increased in PCD mice which also corresponded to an increase in protein production in the cerebellum, but not in the blood plasma. Here we hypothesize that these changes are due to the onset of cell death in the cerebellum, as VEGF-B production occurs after neuronal damage and plays a key role in cell repair. We consider this fluctuation to be local as VEGF-B production occurs in neuronal cells and is therefore unchanged in blood plasma.

On the other hand, at P20 we found a reduction in gene expression in the cerebellum and in protein levels in the blood plasma in mutant mice, without changes in protein production in the cerebellar tissue. Again, these changes may be due to the fact that Vegf-B production occurs mainly in neuronal cells. As P20 is the time at which Purkinje cell death begins, the reduction in Vegf-B gene expression may lead to a general reduction due to such neuronal loss. However, since it is so necessary for its reparative and anti-apoptotic effect, it is possible that the protein produced is maintained in the cerebellum, thus preventing or attempting to prevent neuronal death, since we did not find any changes at the protein level in the cerebellum, but we did find a reduction in blood plasma.

In light of the aforementioned information and in response to the Reviewer's suggestion, we have incorporated this data and enhanced the discussion to facilitate a more comprehensive understanding of the results (lines 556-565).

2, Since rhVEGF-B does not enhance Purkinje cell survival but improves motor function in mice, it is important to assess Purkinje cell morphology, as reported in the following study (PMID: 37082980).

Actually, the treatment with rhVEGF-B did enhance Purkinje cell survival in PCD mice. Our analysis at P25 demonstrated that the treated mice exhibited a comparable Purkinje cell density to WT mice. That is to say, the VEGF-B treatment virtually stopped the neuronal death along one entire week (from P18 to P25) in a model of disease in which the complete degeneration takes only 4 weeks longer. However, at P30, although we found a significantly higher number of Purkinje cells in treated mice compared to PCD mutants, the neuronal density was significantly lower than in WT mice. In this sense, a clear delayed in neuronal death was detected.

We believe that morphological analysis is an excellent method for evaluating the quality of surviving Purkinje cells and we did it following the Reviewer’s suggestion. However, due to the limited time available for the review process, we could conduct just a qualitative analysis and included a supplementary figure comparing Purkinje cells in WT, PCD, PCD-IGF1 and PCD-VEGF-B mice (Figure S3). As it can be seen from these new results, the quality of Purkinje cells in rhVEGF-B treated mice is similar to that of control mice. Conversely, the surviving cells in PCD mice treated with rhIGF-1 are comparable to those in untreated PCD mice, further substantiating the efficacy of the rhVEGF-B and the neutral effect of rhIGF-1 treatment. These results align with those of previous studies conducted in our laboratory, which demonstrated that enhanced cell survival is associated with improved PC morphology, with the increase in cell number always taking precedence over cell morphology (Pérez-Martín, Muñoz-Castañeda et al., 2021).

3, rhVEGF-B reduces Purkinje cell apoptosis levels. What is the potential mechanism? I recommend the authors stain for Caspase-3 or other apoptosis-related markers following treatment to provide more insights.

First of all, we would like to apologize if there has been any misunderstanding about the interpretation of results. This has been corrected in the manuscript and is also described below.

In this work, we have not focused on the specific mechanisms by which rhVEGF-B inhibits apoptosis (see later). However, it is widely studied that VEGF-B has an anti-apoptotic effect, which promotes cell survival. This effect is achieved through its interaction with the VEGFR1 receptor, which we demonstrated that is expressed in neurons, especially in Purkinje cells. This interaction induces the expression of anti-apoptotic genes such as Bcl-2 and inhibits pro-apoptotic genes such as those of the BH3 subfamily. The interaction VEGF-B - VEGFR1 also inhibits cell death-related proteins, including p53 and members of the caspase family (discussion included in the manuscript). Nevertheless, in order to verify the decrease in apoptosis, it would be necessary to analyze other factors, such as pro- and anti-apoptotic proteins in the cerebellum at earlier ages, before the onset of neuronal death (P15, P20), when animals are still being treated with rhVEGF-B and its effect is greater. This specific study of apoptosis is beyond the scope of this paper.

In any case, following the recommendations of the Reviewer, we analyzed the expression of active caspase 3 in the cerebellum of WT, PCD and PCD-rhVEGFB mice at P25 (peak of neurodegeneration in PCD mice and age of maximum effect of rhVEGF-B), in order to ascertain whether it was related to the results obtained in TUNEL. The expression of active caspase 3 serves as an excellent biomarker to monitor the induction of apoptosis, while TUNEL is a method used to analyze apoptosis via DNA fragmentation, which is a more delayed stage of apoptosis. In this regard, analyses performed at P25 have shown a qualitative increase of caspase 3 positive cells in rhVEGF-B treated mice compared to PCD mice. These results are in line with TUNEL labelling at P30 (later, when the apoptotic events are more developed) in PCD-rhVEGFB mice, when we observed an increased number of TUNEL-positive cells compared to PCD. In summary, the results of the study of Purkinje cell viability indicate that at P25 rhVEGF-B-treated mice exhibited a cell density comparable to WT mice. However, at this age, these cells are beginning their apoptotic process (increased caspase 3 labeling), which reached their peak at P30, as indicated by the TUNEL analysis. This new experiment supports our hypothesis that rhVEGF-B treatment delays neuronal death in Purkinje cells, as we observed more apoptosis at later stages, precisely because of the increase in cell viability. We have included this analysis as Supplementary Figure 4.

Furthermore, preliminary data from our laboratory indicate that the mechanism underlying VEGF-B neuroprotection may be associated with alterations in mitochondrial quality. Firstly, it has been observed that there are mitochondrial alterations associated with the pcd mutation. These include changes in mitochondrial quality parameters, such as downregulation of mitochondrial subunits or upregulation of enzymes related to mitochondrial dysfunction, which occur from the onset of neuronal degeneration. Conversely, treatment with rhVEGF-B appears to normalize these levels in addition to oxidative phosphorylation, which may represent the mechanism of Purkinje cell enhancement. This preliminary information suggests that the mechanism of VEGF-B neuroprotection may lie in the enhancement of mitochondrial quality. Nevertheless, further research is required to substantiate this hypothesis. However, since this is not the primary focus of the present study, and since these data are the result of an ongoing study involving several laboratories, they have not been included in the manuscript.

Reviewer 2 Report

Comments and Suggestions for Authors

The manuscript has the following issues:

1. The effects of human recombinant IGF-1 (rhIGF-1) or VEGF-B (rhVEGF-B) treatments on motor, Cognitive and social behavior using the Purkinje Cell Degradation (PCD) mouse. The experimental results in this manuscript seem to lack direct experimental results, such as recorded results or videos. It is recommended to supplement them to provide readers with a more complete understanding of the experimental results.

2. Documents lacking supplementary information.

3. The saturation of immunofluorescence and TUNEL images is weak, and the author should improve them to enhance the quality of these images. Easy for readers to read.

4. In order to facilitate readers' understanding, the author should draw a concise experimental flowchart to improve the readability of the manuscript.

Comments on the Quality of English Language

The English could be improved to more clearly express the research.

Author Response

1. The effects of human recombinant IGF-1 (rhIGF-1) or VEGF-B (rhVEGF-B) treatments on motor, Cognitive and social behavior using the Purkinje Cell Degradation (PCD) mouse. The experimental results in this manuscript seem to lack direct experimental results, such as recorded results or videos. It is recommended to supplement them to provide readers with a more complete understanding of the experimental results.

For the behavioral analyses, approximately 20 videos were analyzed per experimental group, at different ages and treatments, using the Anymaze software. Given the considerable number of videos and tests conducted, we decided to present the results just in graphical form, as a concise and manageable summary of results, and since providing access to all the videos would have made the comprehension of results more challenging than beneficial.

Nevertheless, we understand that the Reviewer wants more visual or raw data, so we have included a heat map of the novel object recognition and social preference tests as illustrative examples of the behavior of the PCD-rhIGF1 and PCD-rhVEGFB experimental groups at P30, which we consider to be the most representative age. In this way, we provide a direct summary of the test's progression, also avoiding the full visualization of the videos, which could be excessively tedious. This information has been added as Supplementary Figure 2 and referenced throughout the manuscript.

2. Documents lacking supplementary information.

We apologize for the inconvenience caused by the lack of access to the full article information. We have uploaded all the files in accordance with the journal's guidelines, but we cannot rule out the possibility of an upload or download issue.

To facilitate the review, we have re-uploaded all the documents to the platform and sent them directly to the Reviewer in PDF format.

3. The saturation of immunofluorescence and TUNEL images is weak, and the author should improve them to enhance the quality of these images. Easy for readers to read.

We decided to utilize TUNEL panoramic images in order to present a comprehensive overview of the extent of cell death in the cerebellum under different experimental conditions. Furthermore, the difficulty in observing the cells may be attributed to the low number of cells in certain genotypes, which is markedly disparate, for instance, between the WT and PCD groups, or even between the PCD and PCD treated with rhVEGF-B at P25.

In any case, in accordance with the Reviewer’s recommendations and to facilitate the comprehension and review of the manuscript, we have modified the images and highlighted the TUNEL-positive cells (Figures 6 and 7).

4. In order to facilitate readers' understanding, the author should draw a concise experimental flowchart to improve the readability of the manuscript.

The flowchart was previously provided as Supplementary Figure 3. However, as stated in point 2 (see above), some troubles should happen with the uploading/downloading process. Then, it has now been re-uploaded with the remaining files and also sent directly to the Reviewer. Taking into account the alterations of the figures during the revision process, the figure number has been changed to Supplementary Figure 6.

Round 2

Reviewer 2 Report

Comments and Suggestions for Authors

The manuscript still has the following issues:

1. The inhibition of cerebellar Purkinje cell degeneration by hrVEGFB should have a dose-response and time-dependent relationship. The author stated in the abstract that hrVEGFB is effective at moderate doses and does not comply with the basic principles of pharmacodynamics. Please provide an explanation.

2. Is the magnification different between Fig7B (PCD group) and Fig7A (WT group), as well as Fig7C (PCD+rhVEGF-B group)?

3. The abbreviations in the entire manuscript should be standardized. For example, for the same factor, there are three abbreviations in the text, hrVEGFB,hrVEGF-B,VEGFB。 Causing difficulties for readers to read, please ask the author to make revisions.

Author Response

  1. The inhibition of cerebellar Purkinje cell degeneration by hrVEGFB should have a dose-response and time-dependent relationship. The author stated in the abstract that hrVEGFB is effective at moderate doses and does not comply with the basic principles of pharmacodynamics. Please provide an explanation.

On this point we do not agree with the Reviewer, as the efficacy of rhVEGF-B treatment adheres to the principles of pharmacodynamics. In this case, we observed an inverted U-shaped dose-response curve, a non-linear relationship, where the effects of increasing doses also appear to increase up to a maximum, and then, at even higher doses, these effects decrease (deleterious effects). This phenomenon, characterized by the high frequency of a particular treatment not being tolerated, and instead, being better at lower frequencies or doses, is a well-documented characteristic of many drugs (Harvey et al., 2022). In the specific case of neurotrophic factors, an excessive dose can sometimes have a detrimental effect on cell development and survival (Miquerol et al., 2000), which may be a possible explanation for our results. Therefore, it is essential to carefully regulate the administration of this type of substance. In this study, the treatment was administered according to the effective dosage established in previous studies (Castro et al., 2014; Yue et al., 2014), adapting it to the timing of Purkinje cell degeneration in PCD mice. In this sense, other pharmacological treatments have shown comparable patterns in the PCD mouse. This is the case of oleylethanolamide, whose chronic treatment did not induce improvements in Purkinje cell density in PCD mice, while more spaced or even single doses were much more effective (Pérez-Martín et al., 2021).

Furthermore, the temporal dynamics of the observed effects suggest a time-dependent nature, with the neuroprotective effect manifesting during the treatment period (P10-P20). These effects seem to last a little longer, since at P25, when the treatment had ended, we still found a clear neuroprotective effect.

However, following the Reviewer's suggestion, we have revised the discussion to make the article more readable.

  1. Is the magnification different between Fig7B (PCD group) and Fig7A (WT group), as well as Fig7C (PCD+rhVEGF-B group)?

Indeed, the magnification is the same between figures 7A-C. It is possible that a visual comparison might raise doubts, as the parasagittal sections of the vermis may not be exactly the same. Also, we cannot rule out any morphological alteration of tissue due to Purkinje cell degeneration in PCD mice. 

  1. The abbreviations in the entire manuscript should be standardized. For example, for the same factor, there are three abbreviations in the text, hrVEGFB,hrVEGF-B,VEGFB。 Causing difficulties for readers to read, please ask the author to make revisions.

Following the Reviewer's suggestions, we have revised the manuscript to check the abbreviations used, to improve reading comprehension. However, we would like to clarify that there are several abbreviations that must be used, even appearing similar or confusing:

  • VEGF-B and Vegf-B denote the protein and gene present in the organism and its receptors. Similarly, we use IGF-1 and Igf-1 to refer to the protein and gene, respectively.
  • Recombinant human VEGF-B (rhVEGF-B) refers to the molecule that we administered to mice. The same applies to rhIGF-1 (recombinant human IGF-1).
  • Finally, to avoid repetitions and to facilitate understanding, we have created the terms PCD-rhIGF1 and PCD-rhVEGFB to refer to those PCD mice treated with the recombinant human VEGF-B (rhVEGF-B) or IGF-1 (rhIGF-1) proteins. This is specified at the start of each treatment. However, we know that this nomenclature is different from rhVEGF-B, as we have removed the hyphen from rhVEGF-B (PCD-rhVEGFB). We did this to avoid multiple hyphens in the same word or abbreviation, and to make it easier to read.

Round 3

Reviewer 2 Report

Comments and Suggestions for Authors

Although the author has answered some questions regarding the round 3 revision, in order to facilitate readers' understanding, please add the relevant content of response 1 in concise sentences to the abstract of the manuscript.

Author Response

Although the author has answered some questions regarding the round 3 revision, in order to facilitate readers' understanding, please add the relevant content of response 1 in concise sentences to the abstract of the manuscript.

Following the Reviewer's suggestion, we have added the information to the abstract.